# Multi-Step Visual Reasoning with Visual Tokens Scaling and Verification

**Tianyi Bai**[1,3*], **Zengjie Hu**[2*], **Fupeng Sun**[4*], **Jiantao Qiu**[3†], **Yizhen Jiang**[2],
**Guangxin He**[1], **Bohan Zeng**[2], **Conghui He**[3‡], **Binhang Yuan**[1‡], **Wentao Zhang**[2‡]
[1]The Hong Kong University of Science and Technology, [2]Peking University
[3]Shanghai Artificial Intelligence Laboratory, [4]Imperial College London
heconghui@pjlab.org.cn, biyuan@ust.hk, wentao.zhang@pku.edu.cn

## Abstract

Multi-modal large language models (MLLMs) have achieved remarkable capabilities by integrating visual perception with language understanding, enabling applications such as image-grounded dialogue, visual question answering, and scientific analysis. However, most MLLMs adopt a static inference paradigm, encoding the entire image into fixed visual tokens upfront, which limits their ability to iteratively refine understanding or adapt to context during inference. This contrasts sharply with human perception, which is dynamic, selective, and feedback-driven. In this work, we introduce a novel framework for inference-time visual token scaling that enables MLLMs to perform iterative, verifier-guided reasoning over visual content. We formulate the problem as a Markov Decision Process, involving a reasoner that proposes visual actions and a verifier—trained via multi-step Direct Preference Optimization (DPO)—that evaluates these actions and determines when reasoning should terminate. To support this, we present a new dataset, VTS, comprising supervised reasoning trajectories (VTS-SFT) and preference-labeled reasoning comparisons (VTS-DPO). Our method significantly outperforms existing approaches across diverse visual reasoning benchmarks, offering not only improved accuracy but also more interpretable and grounded reasoning processes. These results demonstrate the promise of dynamic inference mechanisms for enabling fine-grained, context-aware visual reasoning in next-generation MLLMs. Code and datasets are publicly released at https://vts-v.github.io/.

## 1   Introduction

Multi-modal large language models (MLLMs) that can perceive and reason over visual content are a foundational component of modern AI systems. By extending large language models (LLMs) with visual perception capabilities, MLLMs support a wide range of applications—from image-grounded dialogue and visual question answering to robotics and scientific analysis. Yet despite their impressive generalization, one fundamental challenge remains unsolved: *how can we conduct effective inference-time scaling for MLLMs to enable fine-grained, context-aware interaction with visual information*?

Current MLLMs typically adopt a *static inference paradigm*—processing the whole image into a static fixed set of visual tokens in a single step, and conducting all reasoning based solely on this static embedding. This approach limits the model's ability to recover from ambiguity, occlusion, or missing

---

*Equal contribution.

†Project leader.

‡Corresponding authors.

39th Conference on Neural Information Processing Systems (NeurIPS 2025).

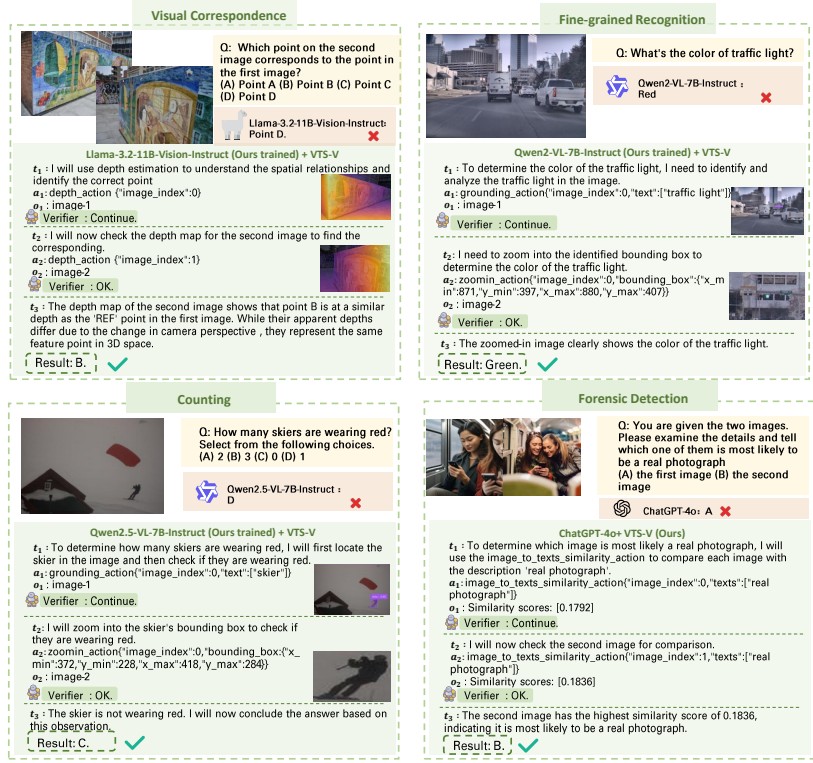

Figure 1: **Iterative Visual Reasoning with VTS-V.** Our framework equips both open-source and closed-source models with dynamic visual token scaling and step-wise verification to solve complex visual tasks. The example shows how VTS-V: (1) decomposes questions into executable steps, (2) invokes vision tools, and (3) iteratively refines answers via verifier feedback, achieving correct results. In contrast, vanilla models fail to ground detailed visual operations without token scaling, leading to incorrect answers.

detail: once the initial representation is formed, there is no mechanism to query the image again or refine visual understanding. In contrast, *human perception is inherently dynamic*—we iteratively inspect regions, zoom in, and seek new visual evidence as reasoning unfolds. Bridging this gap between static MLLM inference and dynamic human reasoning is the central problem addressed in this work.

Multi-step visual reasoning is critical for robust AI systems. Many tasks require identifying small objects, interpreting text in images, or reasoning about spatial relations—activities that benefit from an iterative, context-sensitive exploration of visual content. Furthermore, dynamic scaling enables more efficient reasoning by focusing computational resources on the most relevant parts of the image, rather than uniformly processing all pixels or tokens. Without this flexibility, existing models exhibit degraded performance on benchmarks such as BLINK [6], V-Star Bench [27], and MMVP [23], which are designed to probe deeper visual understanding.

The core technical challenge lies in the absence of an expressive framework for flexible, inference-time visual exploration. Existing approaches either construct improved visual reasoning datasets for fine-tuning [7, 21], which still rely heavily on the pretrained model's image-text alignment quality, or attempt to enhance inference through text token scaling [2] and the use of external vision tools [11, 8, 22, 27]. However, these methods are limited in scope, focusing narrowly on static token expansion or employing a small set of fixed visual tools in predefined pipelines. As a result, these methods fail to give the model agency in choosing what to observe next, how to focus visual attention, or when to stop reasoning. What's needed is a unified framework that allows the model to take structured, interpretable visual actions—guided by feedback—while remaining grounded in the image content.

**Contributions.** To address this, we propose a novel framework for inference-time visual token scaling, enabling MLLMs to engage in *iterative, verifier-guided reasoning* over images. Our contributions are threefold:

- **Expressive and theoretically grounded framework.** We formulate visual reasoning as a Markov Decision Process (MDP) with two key components: a *reasoner* that proposes visual actions, and a *verifier*, trained via multi-step Direct Preference Optimization (DPO), which evaluates action quality and terminates reasoning when appropriate. We prove that our reasoner and verifier cooperation system ensures alignment between reasoning actions and visual content, while guaranteeing a bounded number of steps through an early stopping mechanism.

- **A new dataset for tool-augmented visual reasoning.** We introduce a two-part dataset: *VTS-SFT* (supervised reasoning trajectories with tool use), and *VTS-DPO* (preference-labeled reasoning pairs). These resources enable effective training of both the reasoner and verifier, supporting dynamic visual interaction with multi-step grounding.

- **Comprehensive evaluation and state-of-the-art results.** Our experiments span a variety of vision-language tasks demanding multi-step reasoning. Across these scenarios, our approach significantly outperforms strong baselines, including models augmented with limited tool use and those employing chain-of-thought prompting. We observe not only accuracy improvements but also more interpretable reasoning traces, as the model's step-by-step process explicitly justifies each answer with visual evidence. These results underscore the effectiveness of inference-time visual token scaling: by enabling an AI to look deeper into images in a controlled, stepwise fashion, we achieve new state-of-the-art performance on tasks that previously stymied conventional MLLMs.

These contributions move us toward flexible, grounded, and interpretable visual reasoning in MLLMs—crucial for building AI systems that can "think with their eyes."

## 2 Related Work

**Visual reasoning.** Visual reasoning in VLMs focuses on integrating visual and textual inputs to enable effective decision-making. Early approaches, such as Shikra [2], applied Chain of Thought (CoT)[26] techniques to visual tasks. Meanwhile, methods like SoM[29] and Scaffolding [13] improved reasoning by leveraging visual anchors, such as segmentation maps. V* [27] introduced a two-step CoT approach for high-resolution visual search, demonstrating the potential of structured reasoning in visual contexts. More recently, efforts have been made to develop improved reasoning datasets for training visual language models (VLMs) [21, 7]. However, most of this work primarily focuses on scaling text tokens during the visual reasoning phase. Limited attention has been given to frameworks that scale visual tokens to address complex visual reasoning tasks.

**Visual programming and tool-using.** Recent research has focused on integrating visual tools with large language models (LLMs) and vision-language models (VLMs) to address complex visual tasks. These studies aim to leverage LLMs and VLMs to generate code that utilizes external vision tools for solving complex vision problems [11, 8, 22, 31]. For instance, Visprog [8] and ViperGPT [22] prompt VLMs to produce single-step Python code that interacts with external vision tools, while Visual Sketchpad [10] enhances these methods by introducing multi-step reasoning capabilities, enabling VLMs to rethink and correct execution errors. However, these approaches fail to address the critical aspect of iterative visual reasoning, as their reasoning steps are typically limited to only 1–2 steps. This prevents them from performing reflective refinements akin to VLMs, hindering their ability to tackle complex visual tasks that require deeper inference and multi-step adjustments.

**Verifier design.** Recent advancements have leveraged verifiers to enhance language model reasoning and solution quality. Approaches include deriving reward signals for reasoning [15, 16], combining solution- and step-level verifiers for math problems [33], and using graph search or Monte Carlo rollouts for rationale generation [19, 24]. Training methods range from human annotations for RL with feedback [34] to synthetic data for RL with AI feedback [30]. Some treat verifiers as generative models, scoring solutions via control tokens [12] or likelihoods [17]. Closely related is V-STaR [9], which uses Direct Preference Optimization (DPO) for solution ranking. However, existing verifiers are typically designed for specific tasks and tailored training datasets. Our work utilizes multi-step DPO as a verification mechanism, supported by theoretical guarantees.

## 3 Visual Tokens Scaling with Verification

We first formally formulate the visual reasoning tasks with visual token scaling and verification.

### 3.1 Problem Formulation

At the first step, a question-image pair $s_1$[4] is sampled from some distribution $\mathcal{D}$ as the initial state. For each step $h \geq 2$, we conduct:

- **Planning**: the VLM observes the current state $s_h$, which is the history of the previous $h-1$ reasoning steps, i.e., $s_h = (s_{h-1}, t_{h-1}, a_{h-1}, o_{h-1})$, and generates planning of $h$-th step $t_h$ by some distribution $p(\cdot \mid s_h)$. $t_h$ can be regarded as the text tokens and determines how to manipulate the image at the $h$-th step.

- **Action**: based on planning $t_h$, the VLM further chooses an action $a_h \in \mathcal{A}$ according to policy $\pi(\cdot \mid t_h)$, where $\mathcal{A}$ is the finite action (module) set. This means that the VLM decides which visual module to implement for a specific instruction $t_h$.

- **Observation**: in response to the action, the environment then returns a visual observation $o_h = f_{a_h}(t_h)$ for some deterministic function $f$. Here the visual observation is assumed to be deterministic since it is the code execution result by some visual module, the depth map of an image, for example.

Then we transit to a new state $s_{h+1}$ and a new reasoning step begins.

- **Verification**: after each set of planning, action and observation, a verifier $r^*$ is designed to decide whether to continue this iterative reasoning, or to stop and give out the final result.

We regard the final planning as the final result of the initial question-image pair. Eventually, we will collect a reasoning sequence

$$\tau = (s_1, t_1, a_1, o_1, \ldots, t_{H_\tau}, a_{H_\tau}, o_{H_\tau}, t_{H_\tau+1}) = s_{H_\tau+1} \cup \{t_{H_\tau+1}\}, \tag{1}$$

where $H_\tau$ (a random variable) is the length of the total reasoning steps for path $\tau$.

### 3.2 Visual Reasoning with Visual Token Scaling and Verification

To enhance performance in visual reasoning tasks, the reasoning process will consist of two main components: a reasoner capable of generating visual reasoning steps, and a verifier that guides the reasoner in visual token scaling and determines the terminal condition.

**Reasoner**. The reasoner is a pre-trained VLM augmented with plug-and-play modules, which are denoted by $\mathtt{R}_{\theta_0}$, where $\theta_0$ encapsulates all the parameters. The module tools can be off-the-shelf vision models (depth maps, bounding box generation, etc), table operations (add columns, select row, etc), web search engines, and Python functions (plot figures, calculation, etc). The reasoner is supposed to present the reasoning steps in a sequential format as in equation (1), where $p(\cdot \mid s_h) = \mathtt{R}_{\theta_0}(\cdot \mid s_h)$ and $\pi(\cdot \mid t_h) = \mathtt{R}_{\theta_0}(\cdot \mid t_h)$. In practice, the reasoner can be: (i) a model such as GPT-4o; or (ii) an open-source model fine-tuned on self-crafted datasets. In this case, $\theta_0$ can be further updated to $\hat{\theta}_{\text{SFT}}$ by SFT.

**Verifier**. The verifier $r^*$ is a function that maps the reasoning trajectory generated by the reasoner to a real number. At each reasoning step $h$, the reasoner will continue to generate one additional step if the reward difference between step $h$ and step $h+1$ is no less than some predetermined positive threshold $\epsilon$. In this paper, we utilize a verifier derived from multi-step DPO [28]. Specifically, we begin with a base VLM $\mathtt{V}_{\phi_0}$, capable of generating reasoning sequences, where $\phi_0$ is known and encompasses all the parameters. Using multi-step DPO with preference data, $\phi_0$ is further updated to $\hat{\phi}_{\text{SDPO}}$. Consequently, the verifier can be represented by $\mathtt{V}_{\phi_0}$ and $\mathtt{V}_{\hat{\phi}_{\text{SDPO}}}$ jointly.

### 3.3 Reasoner and Verifier Training

The training stage is divided into two parts: fine-tuning the reasoner with SFT and verifier training based on DPO (answer level and multi-step level). Correspondingly, the training dataset consists $\mathcal{D}_{\text{SFT}}$ and $\mathcal{D}_{\text{DPO}}$, we will describe the details of Visual Tokens Scaling SFT and DPO training dataset construction in Section 4.

**Fine-tune the reasoner by SFT**. Suppose we have a base reasoning model $\mathtt{R}_{\theta_0}$.

*SFT training*. Given a Visual Token Scaling SFT training dataset $\mathcal{D}_{\text{SFT}}$, the supervised finetuning process refers to the learning of reasoner $\mathtt{R}_\theta$ through minimizing the following cross-entropy loss

$$\hat{\theta}_{\text{SFT}} := \operatorname{argmin}_\theta \frac{1}{|\mathcal{D}_{\text{SFT}}|} \sum_{\tau \in \mathcal{D}_{\text{SFT}}} \frac{1}{H_\tau + 1} \mathcal{L}_{\text{SFT}}(\mathtt{R}_\theta(\tau), \tau),$$

---

[4]In practice, $s_1$ will also include system prompts.

where $\mathcal{L}_{\text{SFT}}(\cdot, \cdot)$ is defined as

$$\mathcal{L}_{\text{SFT}}\left(\text{R}_\theta\left(\tau\right), \tau\right) = -\sum_{h=1}^{H_\tau+1} \log \text{R}_\theta\left(t_h^\tau \mid s_h^\tau\right) - \sum_{h=1}^{H_\tau} \log \text{R}_\theta\left(a_h^\tau \mid t_h^\tau\right). \tag{2}$$

$\hat{\theta}_{\text{SFT}}$ then can be obtained by the iteration of gradient descent

$$\theta_{k+1} = \theta_k - \alpha \frac{1}{|\mathcal{D}_{\text{SFT}}|} \sum_{\tau \in \mathcal{D}_{\text{SFT}}} \frac{1}{H_\tau + 1} \nabla \mathcal{L}_{\text{SFT}}\left(\text{R}_\theta\left(\tau\right), \tau\right), \tag{3}$$

where $\alpha$ is the learning rate and the initial parameter is $\theta_0$.

**Verifier training by DPO**. We use multi-step DPO to derive the desired verifier that aligns with human preferences, building upon a base VLM $\text{V}_{\phi_0}$, such as LLaVA-v1.5-7B. Suppose we are given preference pairs $\left(s_1, \tau^w, \tau^l\right)$, where $\tau^w$ is preferred trajectory over unpreferred trajectory $\tau^l$.

The key idea of multi-step DPO is to assume that $r^*$ belongs to a family of one-parameter functions $\{r_\phi(\cdot)\}_\phi$, i.e., $r^*(\cdot) = r_{\phi_{\text{SDPO}}}(\cdot)$ for some unknown ground truth parameter $\phi_{\text{SDPO}}$. Specifically, if $r_\phi(\cdot)$ is defined as

$$r_\phi(\tau) = \eta \sum_{h=1}^{H_\tau+1} \log \frac{\text{V}_\phi\left(t_h \mid s_h\right)}{\text{V}_{\phi_0}\left(t_h \mid s_h\right)} + \eta \sum_{h=1}^{H_\tau} \log \frac{\text{V}_\phi\left(a_h \mid t_h\right)}{\text{V}_{\phi_0}\left(a_h \mid t_h\right)} + Q(s_1),$$

where $\eta$ is a positive constant and $Q(\cdot)$ is a fixed function that depends only on $s_1$, then the preference model $\text{V}_{\phi_{\text{SDPO}}}$ will align with the verifier $r_{\phi_{\text{SDPO}}}$ while remaining close to the original $\text{V}_{\phi_0}$ automatically, as shown by Equation 8 and Proposition A.1.

Furthermore, $\phi_{\text{SDPO}}$ can be determined by maximizing the expected likelihood under the Bradley-Terry model

$$\mathbb{E}_{(s_1, \tau^w, \tau^l)}\left[\mathbb{P}\left(\tau^w \succ \tau^l\right)\right], \tag{4}$$

where

$$\mathbb{P}\left(\tau^w \succ \tau^l\right) = \sigma\left(r^*\left(\tau^w\right) - r^*\left(\tau^l\right)\right). \tag{5}$$

Once $\phi_{\text{SDPO}}$ is obtained, the verifier $r_{\phi_{\text{SDPO}}}$ can be used to guide the reasoner in generating the reasoning procedure. The detailed training procedure is as follows.

*Multi-step DPO*. Define the empirical multi-step DPO loss $\mathcal{L}_{\text{SDPO}}\left(\phi, \phi_0\right)$ by equation (11). Let

$$\hat{\phi}_{\text{SDPO}} = \text{argmin}_\phi \mathcal{L}_{\text{SDPO}}\left(\phi, \phi_0\right),$$

then $\hat{\phi}_{\text{SDPO}}$ can be obtained by the iteration of gradient descent

$$\phi_{k+1} = \phi_k - \alpha' \sum_{(s_1, \tau^w, \tau^l) \in \mathcal{D}_{\text{DPO}}} \nabla \mathcal{L}_{\text{SDPO}}\left(\phi, \phi_0\right), \tag{6}$$

where $\alpha'$ is the learning rate and the initial parameter is $\phi_0$. The verifier $r^*$ operates on some reasoning trajectory $\tau$ could be approximated by $r_{\hat{\phi}_{\text{SDPO}}}(\tau)$.

### 3.4 Inference Algorithm with Practical Efficiency and Theoretical Guarantees

In this subsection, we show our algorithm during inference time and the theoritical guarantees.

Given a new test pair $s_1 \sim \mathcal{D}$, we use the reasoner $\text{R}_{\hat{\theta}_{\text{SFT}}}$ for scaling the inference sequence, and we use the verifier $\text{V}_{\hat{\phi}_{\text{SDPO}}}$ to instruct reasoner whether to continue generating the reasoning sequence. Assume a reasoning sequence $s_h$ is generated by the reasoner $\text{R}_{\hat{\theta}_{\text{SFT}}}$. We determine whether $\text{R}_{\hat{\theta}_{\text{SFT}}}$ need to generate one additional reasoning step

$$\begin{aligned} t_h &\sim \text{R}_{\hat{\theta}_{\text{SFT}}}\left(\cdot \mid s_h\right), \quad a_h \sim \text{R}_{\hat{\theta}_{\text{SFT}}}\left(\cdot \mid t_h\right), \quad o_h = f_{a_h}\left(t_h\right), \\ s_{h+1} &= \left(s_h, t_h, a_h, o_h\right), \end{aligned} \tag{7}$$

by checking whether $\left|r_{\hat{\phi}_{\text{SDPO}}}\left(s_{h+1}\right) - r_{\hat{\phi}_{\text{SDPO}}}\left(s_h\right)\right| \geq \epsilon$, where $\epsilon$ is some predetermined positive threshold. We repeat such procedure unless $\left|r_{\hat{\phi}_{\text{SDPO}}}\left(s_{h+1}\right) - r_{\hat{\phi}_{\text{SDPO}}}\left(s_h\right)\right| < \epsilon$. The following lemma gives an equivalent explicit termination condition.

**Lemma 3.1.** *The reasoner $R_{\theta_0}$ stops at the reasoning step $h$ if*

$$\left| r_{\hat{\phi}_{SDPO}}(s_{h+1}) - r_{\hat{\phi}_{SDPO}}(s_h) \right| < \epsilon \Leftrightarrow \left| \log \frac{V_{\hat{\phi}_{SDPO}}(t_h \mid s_h)}{V_{\phi_0}(t_h \mid s_h)} + \log \frac{V_{\hat{\phi}_{SDPO}}(a_h \mid t_h)}{V_{\phi_0}(a_h \mid t_h)} \right| < \epsilon.$$

Algorithm 1 characterizes the full training and test procedure of our method.

Given $s_1 \sim \mathcal{D}$, denote by $H_\tau\left(\hat{\phi}_{SDPO}, \phi_0; \hat{\theta}_{SFT}\right)$[5] the total reasoning step given by Algorithm 1, the next theorem shows that $H_\tau\left(\hat{\phi}_{SDPO}, \phi_0; \hat{\theta}_{SFT}\right)$ is finite almost surely (a.s.).

**Theorem 3.2** (Reasoning steps characterization)**.** *The total reasoning step satisfies that*

1. *$H_\tau\left(\hat{\phi}_{SDPO}, \phi_0; \hat{\theta}_{SFT}\right)$ is a stopping time.*

2. *(Informal). Under some mild condition, $H_\tau\left(\hat{\phi}_{SDPO}, \phi_0; \hat{\theta}_{SFT}\right)$ is finite with probability 1.*

Theorem 3.2 shows that our algorithm is not only flexible—allowing variational reasoning steps that improve upon the results of Xiong et al. [28]—but also guarantees a bounded number of reasoning steps. This ensures that our method avoids scenarios in which the reasoner would otherwise produce looping or repetitive outputs. The detailed proof can be found in Appendix A.4. The superiority of our algorithm can be shown in the following experiments.

## 4 Visual Tokens Scaling Dataset Construction

As discussed in Section 3, our goal is to train a *reasoner* capable of performing visual token scaling—namely, learning to invoke visual understanding tools throughout the reasoning process to generate intermediate images rich in detailed visual token information. At the same time, we aim to train a *verifier* that can distinguish between two reasoning trajectories of similar structure for the same task, identifying the one that more effectively scales visual information. This collaborative setup enables the verifier to guide and refine the reasoner's visual reasoning process.

However, a major challenge lies in the absence of existing datasets that contain long-chain, tool-grounded reasoning traces with rich intermediate visual states. In particular, no existing datasets provide both detailed visual CoT-style supervision and curated accepted–rejected trajectory pairs required for DPO training. To address this, we construct a dedicated dataset by building upon the single-image dataset of the LLaVA-OneVision dataset (LLaVA-OV) [14], which contains 3.2M vision-language examples covering a broad range of tasks.

In this section, we describe our dataset construction pipeline in detail. Section 4.1 introduces how we collect and preprocess visual reasoning examples from LLaVA-OV to produce structured tool-based trajectories. Section 4.2 then explains how we derive both the SFT and DPO datasets from these verified trajectories.

### 4.1 Visual Token Scaling Dataset Collection and Preprocessing

We begin our dataset construction by sampling image-question pairs from the single-image portion of the LLaVA-OV dataset, which covers a wide range of vision-language tasks such as grounding, chart understanding, OCR, and mathematical reasoning. To ensure sufficient task diversity, we uniformly sample across all 83 task types included in the dataset.

For each sampled example, we employ `Qwen-2.5-VL-72B` in a few-shot prompting setting to generate a visual token scaling trajectory. This trajectory is composed of a sequence of tool-based operations drawn from our predefined action space $\mathcal{A}$ (as described in Section 3.1). In line with prior work [8, 11], the action space comprises ten visual tools: `GroundingAction`, `DepthAction`, `ZoomInAction`, `VisualSearchAction`, `Crop`, `OCR`, `ImageSegment`, `ImageCaptioner`, `SimilarityComputing`, and `Overlay`. These tools allow the model to iteratively perceive, transform, and enrich the visual content throughout the multi-step reasoning process.

Generated trajectories may include errors, invalid tool use, or inconsistent reasoning. We apply a preprocessing pipeline to flatten nested structures, merge related turns, remove malformed examples, and ensure metadata consistency. We also filter out trivial cases with no tool use and those exceeding 20,000 tokens due to memory limits. To guarantee the correctness of each trajectory, we employ

---

[5]For notation simplicity, here we omit the dependence on $s_1$.

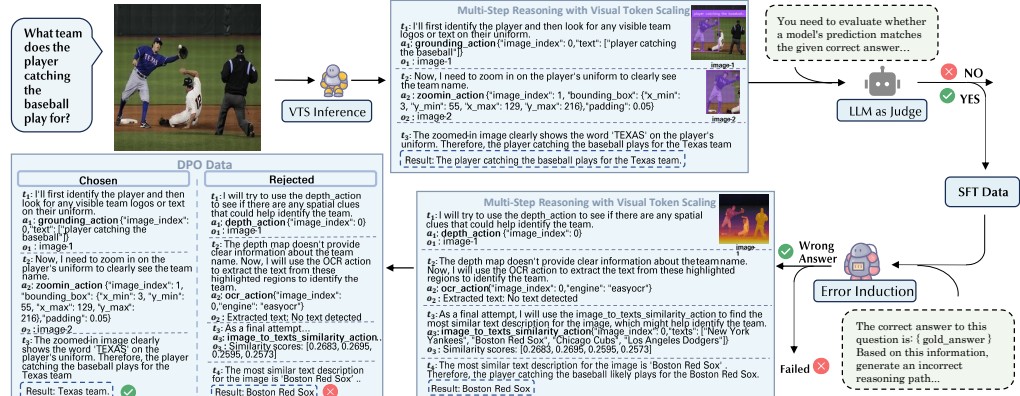

Figure 2: **Pipeline for Synthetic Data Generation and Curation in VTS-V**. Our data construction process consists of three stages: (1) generating multi-step reasoning trajectories with visual tool calls, (2) filtering out incorrect trajectories using an LLM-as-a-judge framework, and (3) creating contrastive (correct vs. incorrect) trajectory pairs for multi-step DPO training.

the same `Qwen-2.5-VL-72B` model as an LLM-based verifier (`llm_as_a_judge`) to filter out those whose final answers are deemed incorrect. Starting from over 650K generated trajectories, this process results in a curated set of 315K high-quality visual token scaling examples, which serve as the basis for supervised fine-tuning and preference training.

## 4.2 VTS-SFT and VTS-DPO Dataset Construction

Starting from the 315K verified visual token scaling trajectories described above, we construct two datasets: one for supervised fine-tuning (VTS-SFT) and one for direct preference optimization (VTS-DPO).

**VTS-SFT Dataset Construction.** To construct the VTS-SFT dataset, we transform each verified trajectory into a supervised training instance. Each trajectory $\tau$ follows the reasoning path format defined in Equation 1, consisting of a sequence of textual states, tool actions, and visual observations, concluding with a final answer. We retain trajectories where the final answer satisfies $t_{H_\tau+1} = t^*$, where $t^*$ is the ground-truth label associated with the input $s_1$. The resulting supervised dataset is defined as:

$$\mathcal{D}_{\text{SFT}} = \left\{ (s_1, \tau) \mid t_{H_\tau+1} = t^*, \ \texttt{llm\_as\_a\_judge}(\tau) = \texttt{correct} \right\}.$$

Each trajectory in $\mathcal{D}_{\text{SFT}}$ is tool-grounded and preserves all intermediate reasoning states, providing rich supervision for training. After processing, the dataset contains approximately 315K high-quality examples.

**VTS-DPO Dataset Construction.** To construct the preference dataset $\mathcal{D}_{\text{DPO}}$, we generate suboptimal reasoning trajectories as contrastive pairs. For each $(s_1, \tau^w) \in \mathcal{D}_{\text{SFT}}$, where $\tau^w$ is a correct trajectory, we design prompts that instruct the model to begin from an intermediate point and then proceed with incorrect reasoning steps, yielding a wrong final answer, which is also the Error Induction step in Figure 4. The resulting suboptimal trajectory $\tau^l$ is paired with $\tau^w$ to form a preference tuple $(s_1, \tau^w, \tau^l)$.

We format these pairs into the standard DPO training structure and remove any example where either $\tau^w$ or $\tau^l$ contains empty or missing image inputs. After filtering, the final dataset $\mathcal{D}_{\text{DPO}}$ comprises 301K preference pairs for training the verifier to assess and guide visual token scaling quality.

## 5 Experiments

We conduct comprehensive experiments to evaluate the effectiveness of our proposed Visual Token Scaling with Verification (VTS-V) framework across a range of visual reasoning tasks. This section is organized as follows: In Section 5.1, we describe the experimental setup, including model variants, training configurations, and evaluation benchmarks. Section 5.2 presents our end-to-end experiments, comparing VTS-V to several strong baselines in both closed-source and open-source settings. Finally, in Section 5.3, we conduct ablation studies to analyze the individual contributions of visual token scaling and verifier integration.

| Model | Depth | Spatial | Jigsaw | VisCorr | SemCorr | ArtStyle | Count | FunCorr | Local | MultiV | Refl | Fore | Sim | Avg |
|---|---|---|---|---|---|---|---|---|---|---|---|---|---|---|
| **Qwen2.5-VL-7B-Instruct Variants** | | | | | | | | | | | | | | |
| Qwen2.5VL-7B | 65.32 | 83.22 | 56.67 | 40.12 | 24.46 | 58.97 | 65.00 | 19.23 | 41.80 | 43.61 | 25.37 | 34.85 | 79.26 | 49.07 |
| Qwen2.5VL-7B + VTS-V | 66.13 | 37.76 | 53.33 | 58.72 | 36.69 | 58.97 | 41.18 | 23.85 | 45.08 | 42.86 | 26.12 | 38.64 | 64.44 | 45.67 |
| Ours + VTS-V | 70.97 | 86.01 | 68.67 | 54.44 | 33.81 | 67.52 | 65.83 | 30.00 | 49.18 | 55.64 | 38.06 | 36.36 | 80.00 | **56.65** |
| **Qwen2-VL-7B-Instruct Variants** | | | | | | | | | | | | | | |
| Qwen2VL-7B | 57.26 | 79.72 | 54.00 | 33.72 | 31.65 | 51.28 | 73.33 | 18.46 | 54.10 | 45.11 | 33.58 | 38.64 | 53.33 | 48.01 |
| Qwen2VL-7B + VTS-V | 49.19 | 67.83 | 57.05 | 15.72 | 17.56 | 42.74 | 43.33 | 12.90 | 49.59 | 30.83 | 32.84 | 29.55 | 51.85 | 38.54 |
| Ours + VTS-V | 60.48 | 60.48 | 58.00 | 37.21 | 40.58 | 76.92 | 63.33 | 28.35 | 51.64 | 47.37 | 35.82 | 34.09 | 84.21 | **52.19** |
| **LLaMA-3.2-11B-Vision-Instruct Variants** | | | | | | | | | | | | | | |
| LLaMA3.2-11B | 63.71 | 67.13 | 53.33 | 50.58 | 39.57 | 47.86 | 55.00 | 32.31 | 62.30 | 48.12 | 31.34 | 25.76 | 46.67 | 47.98 |
| LLaMA3.2-11B + VTS-V | - | - | - | - | - | - | - | - | - | - | - | - | - | - |
| Ours + VTS-V | 68.55 | 69.23 | 57.33 | 38.60 | 47.48 | 55.56 | 56.67 | 35.43 | 58.20 | 52.63 | 33.58 | 23.48 | 57.46 | **50.32** |
| **GPT-4o Variants** | | | | | | | | | | | | | | |
| GPT-4o | 74.19 | 69.23 | 55.33 | 75.00 | 53.96 | 82.91 | 49.17 | 40.77 | 59.84 | 59.40 | 37.31 | 79.55 | 72.59 | 62.25 |
| GPT-4o + Sketchpad | 83.90 | 81.10 | 70.70 | 80.80 | 58.30 | 77.19 | 66.70 | 42.10 | 65.40 | 45.60 | 33.10 | 79.00 | 84.20 | 66.78 |
| GPT-4o + CoT | 73.39 | 82.52 | 62.00 | 82.56 | 57.55 | 82.05 | 65.00 | 57.69 | 60.66 | 53.38 | 41.04 | 62.88 | 63.70 | 64.96 |
| GPT-4o + SoM | 68.55 | 76.22 | 49.33 | 83.72 | 52.52 | - | 43.33 | 47.69 | 59.84 | 56.40 | - | - | 63.70 | 60.13 |
| GPT-4o + MMFactory | 80.30 | 81.80 | 75.30 | 85.50 | 58.30 | 83.00 | 61.70 | 55.40 | 59.00 | 60.20 | 35.10 | 84.80 | 75.30 | 68.90 |
| GPT-4o + VTS-V (Ours) | 79.84 | 85.31 | 75.33 | 82.56 | 56.83 | 80.34 | 67.50 | 53.08 | 68.85 | 52.63 | 40.30 | 71.21 | 85.19 | **69.15** |

Table 1: **Model performance on BLINK subtasks.** Each column corresponds to a different visual reasoning task in the BLINK benchmark. Highlighted cells show whether using the proposed VTS-V method improves or degrades the performance compared to the base model.

## 5.1 Experimental Setup

**Models and Baselines.** Our experiments are divided into two main parts: closed-source models and open-source models. For the closed-source setting, we follow prior work and evaluate the effectiveness of VTS-V using GPT-4o, comparing it with four strong baselines: (i) Zero-shot reasoning on GPT-4o; (ii) Chain-of-Thought (CoT) reasoning on GPT-4o; (iii) Visual prompting framework: Set-of-Mark (SoM), and Visual Sketchpad; and (iv): Tool-using framework: MMFactory.

For the open-source setting, we fine-tune three vision-language model: Qwen2-VL-7B-Instruct [25], Qwen2.5-VL-7B-Instruct[1], and LLaMA3.2-Vision-Instruct-11B on our VTS-SFT dataset. We evaluate the performance of VTS-V both before and after fine-tuning to assess its impact. Our verifier model is trained on `Qwen2.5-VL-7B-Instruct` using our VTS-DPO dataset. All SFT experiments are conducted using LLaMA Factory under unified settings, and DPO training is carried out with TRL. Detailed training configurations are provided in Appendix B.

**Evaluation Tasks.** We evaluate on four representative vision-language reasoning benchmarks: BLINK [6], $V^*$Bench [27], MMStar [3], and MathVista [18]. These benchmarks span a diverse set of capabilities, including multi-image perception and reasoning (BLINK), fine-grained understanding of small visual objects (VBench), broad general knowledge QA (MMStar), and visual mathematical reasoning (MathVista). Following [11] and [5], we conduct our main experiments on 13 tasks from BLINK, a challenging and diverse benchmark designed to evaluate fine-grained visual reasoning. These tasks span visual and semantic correspondence, spatial understanding, multi-view reasoning, depth and reflectance estimation, counting, object localization, pattern alignment (e.g., jigsaw), art style comparison, functional and semantic matching, visual similarity, and forensic image detection. Further evaluation details are provided in the Appendix.

## 5.2 End-to-End Experiments

**VTS-V improves performance of closed-source models.** As shown in Table 1, our VTS-V framework brings robust and consistent improvements to GPT-4o on the BLINK benchmark. The significant performance gain, achieved without fine-tuning, demonstrates the inherent effectiveness of our verifier-guided visual token scaling method for enhancing visual reasoning. Specifically, the average performance increases from 62.25% to 69.15% (+6.90), outperforming all other prompting-based and tool-augmented variants. Compared with GPT-4o + MMFactory, VTS-V yields a slight but meaningful gain of +0.25. It shows larger improvements over GPT-4o + CoT (+3.30), GPT-4o + SoM (+9.02), and GPT-4o + Sketchpad (+2.37). The largest gains are observed on complex compositional tasks, such as Counting (+18.33), Functionally-Correlated (+12.91), and Visually-Correlated (+7.56), indicating that VTS-V strengthens reasoning over fine-grained and structured visual information.

**VTS-V enhances open-source models fine-tuned on our dataset.** Open-source models also benefit significantly from our framework. When these models are first fine-tuned on our VTS-SFT dataset and then evaluated using VTS-V, consistent gains are observed. For example, Qwen2VL-7B improves from 48.01% to 52.19% (+4.18), showing gains in Multi-view (+3.44), Local (+2.48), and

Jigsaw (+4.08). Similarly, Qwen2.5VL-7B improves from 49.07% to 56.65% (+7.58), with boosts in Depth (+5.65), Spatial (+2.79), and Visual Correlation (+14.32). LLaMA3.2-11B, despite being a larger model, shows an improvement from 47.98% to 50.32% (+2.34), especially on tasks like Semantic Correlation (+8.89) and Functional Correlation (+3.12). These results demonstrate that VTS-V generalizes well to open-source architectures and consistently improves performance across vision reasoning categories.

**Fine-tuning on VTS-SFT datasets enhances models' tool-using abilities.** We find that applying VTS-V directly to open-source models without supervised fine-tuning can harm performance. For instance, applying VTS-V to Qwen2VL-7B drops performance from 48.01% to 38.54%, and LLaMA3.2-11B fails to output any valid tool-use actions. This behavior contrasts with GPT-4o, which can reliably follow verifier guidance even without additional training. These results suggest that base open-source models do not possess the necessary behavior patterns to interact properly with verifier signals or external tools. However, after supervised fine-tuning on our VTS-SFT dataset, these models not only recover but surpass their original performance. Post-finetuning, Qwen2VL-7B improves to 52.19%, Qwen2.5VL-7B rises to 56.65%, and LLaMA3.2-11B reaches 50.32%. More importantly, they now produce valid tool-use actions, indicating successful alignment with verifier guidance. This confirms that the VTS-SFT dataset is critical for enabling reliable tool interaction in open-source VLMs.

**VTS-V generalizes to diverse benchmarks.** To assess generalization, we evaluate models on V*Bench[27], MMStar[3], and MathVista[18] (Table 2). Without fine-tuning, applying VTS-V can degrade performance—for instance, Qwen2.5VL-7B drops from 73.30 to 67.54 on VBench. After full training (VTS-SFT + VTS-V), performance improves across all models: Qwen2.5VL-7B reaches 75.13, Qwen2VL-7B recovers to 66.67, and LLaMA3.2-11B rises to 60.21. The results for GPT-4o on these benchmarks can be found in Table 5.

| Trained Model | V*Bench | MMStar | MathVista |
|---|---|---|---|
| Qwen2.5VL-7B | 73.30 | 55.00 | 67.50 |
| Qwen2.5VL-7B + VTS-V | 67.54 | 55.93 | 52.60 |
| Qwen2.5VL-7B (Ours) + VTS-V | 75.13 | 57.93 | 66.50 |
| Qwen2VL-7B | 66.49 | 53.93 | 58.20 |
| Qwen2VL-7B + VTS-V | 50.80 | 44.29 | 37.50 |
| Qwen2VL-7B (Ours) + VTS-V | 66.67 | 55.26 | 60.30 |
| LLaMA-3.2-11B | 54.45 | 50.57 | 32.20 |
| LLaMA-3.2-11B + VTS-V | – | – | – |
| LLaMA-3.2-11B (Ours) + VTS-V | 60.21 | 50.71 | 48.10 |

Table 2: **Model performance on general benchmarks.** Evaluation on V*Bench, MMStar, and MathVista. Dashes indicate missing results.

### 5.3 Ablation Study and Discussion

We compare `GPT-4o + VTS-V` with a text-only chain-of-thought baseline (`GPT-4o + CoT`). `VTS-V` achieves 69.15% average accuracy, surpassing CoT's 64.96% by +4.19 points. The gains are especially strong on tasks needing structured visual reasoning, such as Counting (+10.90), Functionally-Correlated (+10.98), and Visually-Correlated (+1.76). These results highlight that scaling visual reasoning via verifier-guided tool use is more effective than relying solely on extended textual reasoning.

**Verifier improves reasoning quality.** Ablation results in Table 3 clearly demonstrate the effectiveness of the verifier module. When the verifier is removed ("w/o verifier"), accuracy consistently drops across all models—Qwen2.5VL sees a decline from 54.44 to 47.09, Qwen2VL from 37.21 to 33.14, and Llama3.2vision from 38.60 to 36.63. This indicates that the verifier plays a crucial role in improving rea-

| | Qwen2.5VL | Qwen2VL | Llama3.2vision |
|---|---|---|---|
| Full | **54.44** | **37.21** | **38.60** |
| w/o verifier | 47.09 | 33.14 | 36.63 |
| w/o VTS-V | 40.70 | 19.77 | 31.98 |

Table 3: Ablation results: accuracy drops when removing verifier or VTS-V.

soning by filtering out suboptimal answer paths and enhancing decision quality. These improvements are even more pronounced compared to removing the whole VTS and verifier, which leads to further substantial accuracy drops.

## 6 Conclusions and Limitations

This paper presents a new framework, Visual Token Scaling with Verification (VTS-V), to enhance multi-step visual reasoning by iteratively selecting visual actions and verifying their utility at each

step. By framing the process as a Markov Decision Process and integrating a verifier trained via step-wise Direct Preference Optimization, the approach enables models to progressively refine their understanding of complex visual inputs. VTS-V achieves strong performance across a range of challenging visual reasoning benchmarks, outperforming existing baselines.

**Limitations.** The framework can be adapted to support a broader set of visual tools and more diverse reasoning formats. Enhancing the verifier's adaptability across tasks and integrating it more tightly with downstream applications may further improve robustness. We believe VTS-V offers a flexible foundation for future research in compositional and tool-augmented visual reasoning.

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

# Appendices

# A Proofs

In this section, we provide the detailed proof that is omitted in the main paper.

## A.1 Verifier Obtaining

Given the base model $V_{\phi_0}$ and preference pairs $(s_1, \tau^w, \tau^l)$, the key idea of multi-step DPO (Rafailov et al. [20]; Xiong et al. [28]) is to assume that $r^*$ has a specific structure under which the new preference model $V_{\phi_{\text{SDPO}}}$ and verifier $r^*$ can be obtained by jointly solving the following KL-regularized planning problem and the maximum likelihood estimation of the Bradley-Terry model.

$$
\phi_{\text{SDPO}} \in \operatorname{argmin}_\phi \mathbb{E}_{H_\tau} \mathbb{E}_{s_1 \sim \mathcal{D}, \{t_h \sim V_\phi(\cdot|s_h)\}_{h=1}^{H_\tau+1}, \{a_h \sim V_\phi(\cdot|t_h), o_h = f_{a_h}(t_h)\}_{h=1}^{H_\tau}} \Bigg[ -r^*(\tau) +
$$
$$
\eta \sum_{h=1}^{H_\tau+1} D_{\text{KL}} \left( V_\phi(\cdot \mid s_h) \| V_{\phi_0}(\cdot \mid s_h) \right) - \eta \sum_{h=1}^{H_\tau} D_{\text{KL}} \left( V_\phi(\cdot \mid t_h) \| V_{\phi_0}(\cdot \mid t_h) \right) \Bigg],
\tag{8}
$$

$$
\phi_{\text{SDPO}} \in \operatorname{argmin}_\phi - \mathbb{E}_{(s_1, \tau^w, \tau^l)} \left[ \mathbb{P} \left( \tau^w \succ \tau^l \right) \right] = \operatorname{argmin}_\phi - \mathbb{E}_{(s_1, \tau^w, \tau^l)} \left[ \sigma \left( r^*(\tau^w) - r^*(\tau^l) \right) \right].
\tag{9}
$$

Firstly, the next proposition shows that problem (8) can be solved directly when $r^*$ lies in a specific family of one-parameter functions,.

**Proposition A.1.** *If $r^*(\tau) = r_{\phi'}(\tau) = \eta \sum_{h=1}^{H_\tau+1} \log \frac{V_{\phi'}(t_h|s_h)}{V_{\phi_0}(t_h|s_h)} + \eta \sum_{h=1}^{H_\tau} \log \frac{V_{\phi'}(a_h|t_h)}{V_{\phi_0}(a_h|t_h)} + Q(s_1)$ for some $\phi'$, where $Q(\cdot)$ is some fixed function that only depends on $s_1$, then the optimal solution of problem (8) is $\phi'$.*

Hence if $r^*$ takes the form $r_{\phi'}$, by Proposition A.1 and equation (8), we have $r^* = r_{\phi_{\text{DPO}}}$. Then to further obtain $\phi_{\text{SDPO}}$, we can plug $r_{\phi'}(\tau)$ in equation (9) and optimize over $\phi'$, i.e.,

$$
\phi_{\text{SDPO}} \in \operatorname{argmin}_{\phi'} - \mathbb{E}_{(s_1, \tau^w, \tau^l)} \left[ \sigma \left( r_{\phi'}^*(\tau^w) - r_{\phi'}^*(\tau^l) \right) \right]
$$
$$
= \operatorname{argmin}_{\phi'} - \mathbb{E}_{(s_1, \tau^w, \tau^l)} \Bigg[ \log \sigma \Bigg( \eta \Bigg( \sum_{h=1}^{H_{\tau^w}+1} \log \frac{V_{\phi'}\left(t_h^{\tau^w} \mid s_h^{\tau^w}\right)}{V_{\phi_0}\left(t_h^{\tau^w} \mid s_h^{\tau^w}\right)} + \sum_{h=1}^{H_{\tau^w}} \log \frac{V_{\phi'}\left(a_h^{\tau^w} \mid t_h^{\tau^w}\right)}{V_{\phi_0}\left(a_h^{\tau^w} \mid t_h^{\tau^w}\right)}
$$
$$
- \sum_{h=1}^{H_{\tau^l}+1} \log \frac{V_{\phi'}\left(t_h^{\tau^l} \mid s_h^{\tau^l}\right)}{V_{\phi_0}\left(t_h^{\tau^l} \mid s_h^{\tau^l}\right)} - \sum_{h=1}^{H_{\tau^l}} \log \frac{V_{\phi'}\left(a_h^{\tau^l} \mid t_h^{\tau^l}\right)}{V_{\phi_0}\left(a_h^{\tau^l} \mid t_h^{\tau^l}\right)} \Bigg) \Bigg) \Bigg].
\tag{10}
$$

Here we use superscript to indicate which trajectory path the reasoning steps lie in. Now we can use the following empirical multi-step DPO loss

$$
\mathcal{L}_{\text{SDPO}}(\phi, \phi_0) = - \sum_{(s_1, \tau^w, \tau^l) \in \mathcal{D}_{\text{DPO}}} \log \sigma \Bigg( \eta \Bigg( \sum_{h=1}^{H_{\tau^w}+1} \log \frac{V_{\phi'}\left(t_h^{\tau^w} \mid s_h^{\tau^w}\right)}{V_{\phi_0}\left(t_h^{\tau^w} \mid s_h^{\tau^w}\right)} + \sum_{h=1}^{H_{\tau^w}} \log \frac{V_{\phi'}\left(a_h^{\tau^w} \mid t_h^{\tau^w}\right)}{V_{\phi_0}\left(a_h^{\tau^w} \mid t_h^{\tau^w}\right)}
$$
$$
- \sum_{h=1}^{H_{\tau^l}+1} \log \frac{V_{\phi'}\left(t_h^{\tau^l} \mid s_h^{\tau^l}\right)}{V_{\phi_0}\left(t_h^{\tau^l} \mid s_h^{\tau^l}\right)} - \sum_{h=1}^{H_{\tau^l}} \log \frac{V_{\phi'}\left(a_h^{\tau^l} \mid t_h^{\tau^l}\right)}{V_{\phi_0}\left(a_h^{\tau^l} \mid t_h^{\tau^l}\right)} \Bigg) \Bigg),
\tag{11}
$$

and obtain the optimal $\hat{\phi}_{\text{SDPO}} \in \operatorname{argmin}_\phi \mathcal{L}_{\text{SDPO}}(\phi, \phi_0)$ by gradient descent. Finally, we use $r_{\hat{\phi}_{\text{SDPO}}}$ to approximate the verifier $r^* = r_{\phi_{\text{SDPO}}}$.

**Proof of Proposition A.1.** The key idea is, given any $s_1$ and $H_\tau$, using backward induction to check the optimality of $\phi'$ for $1 \leq h \leq H_\tau + 1$. For any given $s_h$, define the value function $Q_h(s_h, \phi)$ as

$$Q_h(s_h, \phi) = \mathbb{E}_{\left\{t_{h'} \sim V_\phi(\cdot|s_{h'})\right\}_{h'=h}^{H_\tau+1}, \left\{a_{h'} \sim V_\phi(\cdot|t_{h'}), o_{h'}=f_{a_{h'}}(t_{h'})\right\}_{h'=h}^{H_\tau}} \left[ -\eta \sum_{h'=h}^{H_\tau+1} \log \frac{V_{\phi'}(t_{h'} \mid s_{h'})}{V_{\phi_0}(t_{h'} \mid s_{h'})} \right.$$

$$+ \eta \sum_{h'=h}^{H_\tau+1} D_{\mathrm{KL}} \left( V_\phi(\cdot \mid s_{h'}) || V_{\phi_0}(\cdot \mid s_{h'}) \right)$$

$$\left. + \eta \mathbf{1}\left\{h \leq H_\tau\right\} \sum_{h'=h}^{H_\tau} \left( -\log \frac{V_{\phi'}(a_{h'} \mid t_{h'})}{V_{\phi_0}(a_{h'} \mid t_{h'})} + D_{\mathrm{KL}} \left( V_\phi(\cdot \mid t_{h'}) || V_{\phi_0}(\cdot \mid t_{h'}) \right) \right) \right]$$

$$- \mathbf{1}\left\{h = 1\right\} Q(s_1).$$

Then our goal is to show that

$$\phi' \in \mathrm{argmin}_\phi \mathbb{E}_{H_\tau} \mathbb{E}_{s_1 \sim \mathcal{D}} Q_1(s_1, \phi). \tag{12}$$

Observe that by the construction, $Q_h(\cdot, \phi)$ satisfies the following recurrence formula

$$Q_h(s_h, \phi)$$
$$= \mathbb{E}_{t_h \sim V_\phi(\cdot|s_h), a_h \sim V_\phi(\cdot|t_h), o_h = f_{a_{h+1}}(t_h)} Q_{h+1}(s_{h+1}, \phi) - \mathbf{1}\left\{h = 1\right\} Q(s_1)$$

$$+ \eta \mathbb{E}_{t_h \sim V_\phi(\cdot|s_h), a_h \sim V_\phi(\cdot|t_h), o_h = f_{a_{h+1}}(t_h)} \left[ -\log \frac{V_{\phi'}(t_h \mid s_h)}{V_{\phi_0}(t_h \mid s_h)} + D_{\mathrm{KL}} \left( V_\phi(\cdot \mid s_h) || V_{\phi_0}(\cdot \mid s_h) \right) \right] \tag{13}$$

$$+ \eta \mathbf{1}\left\{h \leq H_\tau\right\} \mathbb{E}_{t_h \sim V_\phi(\cdot|s_h), a_h \sim V_\phi(\cdot|t_h), o_h = f_{a_{h+1}}(t_h)} \left[ -\log \frac{V_{\phi'}(a_h \mid t_h)}{V_{\phi_0}(a_h \mid t_h)} + D_{\mathrm{KL}} \left( V_\phi(\cdot \mid t_h) || V_{\phi_0}(\cdot \mid t_h) \right) \right]. \tag{14}$$

We now use backward induction to show that for any fixed $s_h$, $1 \leq h \leq H_\tau + 1$, $\phi' \in \mathrm{argmin}_\phi Q_h(s_h, \phi)$.

(a). For $h = H_\tau + 1$,

$$Q_{H_\tau+1}(s_{H_\tau+1}, \phi) = \mathbb{E}_{t_{H_\tau+1} \sim V_\phi(\cdot|s_{H_\tau+1})} \left[ -\eta \log \frac{V_{\phi'}(t_{H_\tau+1} \mid s_{H_\tau+1})}{V_{\phi_0}(t_{H_\tau+1} \mid s_{H_\tau+1})} + \eta D_{\mathrm{KL}} \left( V_\phi(\cdot \mid s_{H_\tau+1}) || V_{\phi_0}(\cdot \mid s_{H_\tau+1}) \right) \right].$$

By directly using Lemma A.2, we can conclude that $Q_{H_\tau+1}(s_{H_\tau+1}, \phi) \geq 0$, where the equality holds when $V_\phi(\cdot \mid s_{H_\tau+1}) = V_{\phi'}(\cdot \mid s_{H_\tau+1})$. Hence $\phi' \in \mathrm{argmin}_\phi Q_{H_\tau+1}(s_{H_\tau+1}, \phi)$.

(b). Assume for $h = h'$, $2 \leq h' \leq H_\tau + 1$, $\phi' \in \mathrm{argmin}_\phi Q_{h'}(s_{h'}, \phi)$. Then for $h = h' - 1$,

$$\phi' \in \mathrm{argmin}_\phi \mathbb{E}_{t_{h'-1} \sim V_\phi(\cdot|s_{h'-1}), a_{h'-1} \sim V_\phi(\cdot|t_{h'-1}), o_{h'-1} = f_{a_{h'}}(t_{h'-1})} Q_{h'}(s_{h'}, \phi).$$

To show $\phi' \in \mathrm{argmin}_\phi Q_{h'-1}(s_{h'-1}, \phi)$, by the definition of $Q_{h'-1}(s_{h'-1}, \phi)$, one only needs to show that both term (13) and term (14) obtain the minimum at $\phi'$ when substituting $h$ to $h' - 1$. This can be checked by using conditional expectation and Lemma A.2.

Hence by combining (a) and (b), we have $\phi' \in \mathrm{argmin}_\phi Q_1(s_1, \phi)$, which implies equation (12), as desired. $\square$

**Lemma A.2** (Proposition 7.16 and Theorem 15.3 of Zhang [32]). *Let $U(\cdot)$ be a given function and $p_0(\cdot)$ be a given density function. Then the solution of*

$$\mathrm{argmin}_{p(\cdot)} \mathbb{E}_{x \sim p(\cdot)} \left[ -U(x) + \eta D_{KL}(p(\cdot) || p_0(\cdot)) \right]$$

*is given by*

$$p^*(x) = \frac{1}{C} p_0(x) \exp\left( \frac{U(x)}{\eta} \right),$$

*where $C = \log \mathbb{E}_{x \sim p_0(\cdot)} \exp\left( \frac{U(x)}{\eta} \right)$. $p^*(\cdot)$ is known as the Gibbs distribution.*

## A.2 Proof of Lemma 3.1

*Proof of Lemma 3.1.* This is directly obtained from the definition of $r_{\hat{\phi}_{\text{SDPO}}}$. $\qquad\square$

## A.3 Practical Algorithm for VTS

---

**Algorithm 1** Visual Reasoning with Visual Token Scaling and Verification

---

**Require** Training dataset $\mathcal{D}_{\text{train}}$, initial reasoner $\text{R}_{\theta_0}$ and base model $\text{V}_{\phi_0}$.
  **Input:** A new test question-image pair $s_1 \sim \mathcal{D}$.
  **Output:** Reasoning trajectory $\tau$.
  1: **Reasoning Sequence Generation:** Initialize reasoning trajectory $\tau \leftarrow \{s_1\}$.
  2: Set step counter $h \leftarrow 1$.
  3: **while** True **do**
  4:     **Planning:** Generate step planning $t_h \sim \text{R}_{\hat{\theta}_{\text{SFT}}}(\cdot \mid s_h)$.
  5:     **Action:** Select action $a_h \sim \text{R}_{\hat{\theta}_{\text{SFT}}}(\cdot \mid t_h)$.
  6:     **Observation:** Return $o_h = f_{a_h}(t_h)$.
  7:     Update State: $s_{h+1} \leftarrow (s_h, t_h, a_h, o_h)$.
  8:     Append $(t_h, a_h, o_h)$ to reasoning trajectory $\tau$.
  9:     **Verification:** Compute reward difference $\Delta r = r_{\hat{\phi}_{\text{SDPO}}}(s_{h+1}) - r_{\hat{\phi}_{\text{SDPO}}}(s_h)$.
 10:     **if** $\Delta r < \epsilon$ **then**
 11:       Break loop.
 12:     **end if**
 13:     $h \leftarrow h + 1$.
 14: **end while**
 15: Generate the final result $t_{H_\tau(\hat{\phi}_{\text{SDPO}})+1} \sim \text{R}_{\hat{\theta}_{\text{SFT}}}\left(\cdot \mid s_{H_\tau(\hat{\phi}_{\text{SDPO}})+1}\right)$, where $H_\tau\left(\hat{\phi}_{\text{SDPO}}, \phi_0; \hat{\theta}_{\text{SFT}}\right)$
    is the total reasoning step.
 16: Return final trajectory $\tau = s_{H_\tau(\hat{\phi}_{\text{SDPO}})+1} \cup \left\{t_{H_\tau(\hat{\phi}_{\text{SDPO}})+1}\right\}$.

---

## A.4 Proof of Theorem 3.2

Before we proceed with the main proof, we provide some relevant definitions and conclusions at the beginning.

**Definition A.3** (Martingale, supermartingale, and submartingale)**.** Let $\mathcal{F}_n$ be a filtration, i.e., an increasing sequence of $\sigma$-fields. A sequence $X_n$ is said to be adapted to $\mathcal{F}_n$ if $X_n \in \mathcal{F}_n$ for all $n$. If $X_n$ is a sequence with

1. $\mathbb{E}|X_n| < \infty$,

2. $X_n$ is adapted to $\mathcal{F}_n$,

3. $\mathbb{E}\left[X_{n+1} \mid \mathcal{F}_n\right] = X_n$ for all $n$,

then $X_n$ is said to be a martingale (with respect to $\mathcal{F}_n$). If in the last definition, $=$ is replaced by $\leq$ or $\geq$, then $X_n$ is said to be a supermartingale or submartingale, respectively.

**Definition A.4** (Stopping time)**.** A random variable $N$ is said to be a stopping time if $\{N = n\} \in \mathcal{F}_n$ for all $n < \infty$, i.e., the decision to stop at time $n$ must be measurable with respect to the information known at that time.

**Theorem A.5** (Martingale convergence theorem, Theorem 4.2.11 of Durrett [4])**.** *If $X_n$ is a submartingale with* $\sup \mathbb{E}X_n^+ < \infty$, *then as $n \to \infty$, $X_n$ converges a.s. to a limit $X$ with $\mathbb{E}|X| < \infty$.*

**Theorem A.6** (Formal version of Theorem 3.2)**.**

1. $H_\tau\left(\hat{\phi}_{SDPO}, \phi_0; \hat{\theta}_{SFT}\right)$ *is a stopping time.*

2. *Assume the following two conditions hold.*

    *(i)* $\sup \mathbb{E}_{\tau_h \setminus \{s_1\} \sim \text{R}_{\hat{\theta}_{SFT}}(\cdot|s_1)} r_{\hat{\phi}_{SDPO}}^+(\tau_h) < \infty$ *for any given $s_1 \sim \mathcal{D}$,*

*(ii) for any $s_{h+1}$, $R_{\hat{\theta}_{SFT}}$, $V_{\phi_0}$ and $V_{\hat{\phi}_{SDPO}}$ satisfy that*

$$D_{KL}\left(R_{\hat{\theta}_{SFT}}\left(\cdot \mid s_h\right) \| V_{\phi_0}\left(\cdot \mid s_h\right)\right) \geq D_{KL}\left(R_{\hat{\theta}_{SFT}}\left(\cdot \mid s_h\right) \| V_{\hat{\phi}_{SDPO}}\left(\cdot \mid s_h\right)\right),$$

*and*

$$D_{KL}\left(R_{\hat{\theta}_{SFT}}\left(\cdot \mid t_h\right) \| V_{\phi_0}\left(\cdot \mid t_h\right)\right) \geq D_{KL}\left(R_{\hat{\theta}_{SFT}}\left(\cdot \mid t_h\right) \| V_{\hat{\phi}_{SDPO}}\left(\cdot \mid t_h\right)\right)$$

.

*Then $H_\tau\left(\hat{\phi}_{SDPO}, \phi_0; \hat{\theta}_{SFT}\right)$ is finite with probability 1.*

***Remark*** A.7 (*Discussion of the conditions in Theorem A.6*). Condition (i) assumes that the verifier is finite for paths generated by reasoner $R_{\hat{\theta}_{SFT}}$ in expectation. This can easily achieved when the verifier is finite.

Condition (ii) can be interpreted as assuming that the distance between $R_{\hat{\theta}_{SFT}}$ and $V_{\phi_0}$ is greater than the distance between $R_{\hat{\theta}_{SFT}}$ and $V_{\hat{\phi}_{SDPO}}$. This assumption is reasonable, as both $R_{\hat{\theta}_{SFT}}$ and $V_{\hat{\phi}_{SDPO}}$ align with human preferences and therefore tend to be closer to each other.

***Proof of Theorem A.6.*** 1. Let $\mathcal{F}_h$ be the smallest $\sigma$-field generated by $s_{h+1}$, i.e., $\mathcal{F}_h = \sigma\left(s_{h+1}\right)$. Observe that by the definition of the stopping rule as shown in equation (8), $\left\{H_\tau\left(\hat{\phi}_{SDPO}, \phi_0; \hat{\theta}_{SFT}\right) = h\right\}$ is determined by $s_{h+1}$. Hence $\left\{H_\tau\left(\hat{\phi}_{SDPO}, \phi_0; \hat{\theta}_{SFT}\right) = h\right\} \in \mathcal{F}_h$, which implies that $H_\tau\left(\hat{\phi}_{SDPO}, \phi_0; \hat{\theta}_{SFT}\right)$ is a stopping time.

2. By the definition of $r_{\hat{\phi}_{SDPO}}$, we have

$$\mathbb{E}\left[r_{\hat{\phi}_{SDPO}}(s_{h+1}) \mid s_h\right]$$

$$= r_{\hat{\phi}_{SDPO}}(s_h) + \mathbb{E}_{t_h \sim R_{\hat{\theta}_{SFT}}(\cdot \mid s_h), a_h \sim R_{\hat{\theta}_{SFT}}(\cdot \mid t_h), o_h = f_{a_h}(t_h)} \left[\log \frac{V_{\hat{\phi}_{SDPO}}(t_h \mid s_h)}{V_{\phi_0}(t_h \mid s_h)} + \log \frac{V_{\hat{\phi}_{SDPO}}(a_h \mid t_h)}{V_{\phi_0}(a_h \mid t_h)}\right].$$

Observe that condition (ii) implies that

$$\mathbb{E}_{t_h \sim R_{\hat{\theta}_{SFT}}(\cdot \mid s_h)}\left[\log \frac{V_{\hat{\phi}_{SDPO}}(t_h \mid s_h)}{V_{\phi_0}(t_h \mid s_h)}\right]$$

$$= D_{KL}\left(R_{\hat{\theta}_{SFT}}\left(\cdot \mid s_h\right) \| V_{\phi_0}\left(\cdot \mid s_h\right)\right) - D_{KL}\left(R_{\hat{\theta}_{SFT}}\left(\cdot \mid s_h\right) \| V_{\hat{\phi}_{SDPO}}\left(\cdot \mid s_h\right)\right)$$

$$\geq 0,$$

and

$$\mathbb{E}_{a_h \sim R_{\hat{\theta}_{SFT}}(\cdot \mid t_h)}\left[\log \frac{V_{\hat{\phi}_{SDPO}}(a_h \mid t_h)}{V_{\phi_0}(a_h \mid t_h)}\right]$$

$$= \mathbb{E}_{a_h \sim R_{\hat{\theta}_{SFT}}(\cdot \mid t_h)}\left[\log \frac{V_{\hat{\phi}_{SDPO}}(a_h \mid t_h)}{R_{\hat{\theta}_{SFT}}(a_h \mid t_h)}\right] + \mathbb{E}_{a_h \sim R_{\hat{\theta}_{SFT}}(\cdot \mid t_h)}\left[\log \frac{R_{\hat{\theta}_{SFT}}(a_h \mid t_h)}{V_{\phi_0}(a_h \mid t_h)}\right]$$

$$= D_{KL}\left(R_{\hat{\theta}_{SFT}}\left(\cdot \mid t_h\right) \| V_{\phi_0}\left(\cdot \mid t_h\right)\right) - D_{KL}\left(R_{\hat{\theta}_{SFT}}\left(\cdot \mid t_h\right) \| V_{\hat{\phi}_{SDPO}}\left(\cdot \mid t_h\right)\right)$$

$$\geq 0,$$

hence $\mathbb{E}\left[r_{\hat{\phi}_{SDPO}}(s_{h+1}) \mid s_h\right] \geq r_{\hat{\phi}_{SDPO}}(s_h)$, which indicates that $r_{\hat{\phi}_{SDPO}}(s_h)$ is a submartingale.

Then combining condition (i) and Martingale convergence theorem A.5, $r_{\hat{\phi}_{SDPO}}(s_h)$ converges a.s. to a limit $\tilde{r}(s_1)$ as $h \to \infty$. Hence for $\epsilon > 0$, with probability 1, there exists a $H_\epsilon(s_1)$ such that $\left|r_{\hat{\phi}_{SDPO}}(s_{h+1}) - r_{\hat{\phi}_{SDPO}}(s_h)\right| < \epsilon$ when $h > H_\epsilon(s_1)$. Therefore, $H_\tau\left(\hat{\phi}_{SDPO}, \phi_0; \hat{\theta}_{SFT}\right) \leq H_\epsilon(s_1)$ with probability 1, as desired.

$\square$

# B   Experimental Details

Table 4: **Hyperparameters for training Qwen2-VL-7B-Instruct & Qwen2.5-VL-7B-Instruct & LLaMA-3.2-11B-Vision-Instruct models**

| Hyperparameter | Value |
|---|---|
| LoRA Rank | 8 |
| LoRA $\alpha$ | 16 |
| LoRA Dropout | 0 |
| LoRA Target | all |
| GPU | $8 \times$ NVIDIA A800 |
| Per Device Train Batch Size | 1 |
| Gradient Accumulation Steps | 1 |
| Warmup Ratio | 0.03 |
| Learning Rate | 3e-5 |
| Learning Rate Scheduler | Cosine |
| Unfreeze Vision Tower | False |
| Number Train Epoch | 1 |
| Max Gradient Norm | 1.0 |
| bf16 | True |
| Cut Off Length | 65536 |

# C   Examples

## C.1   VTS-DPO Dataset Examples

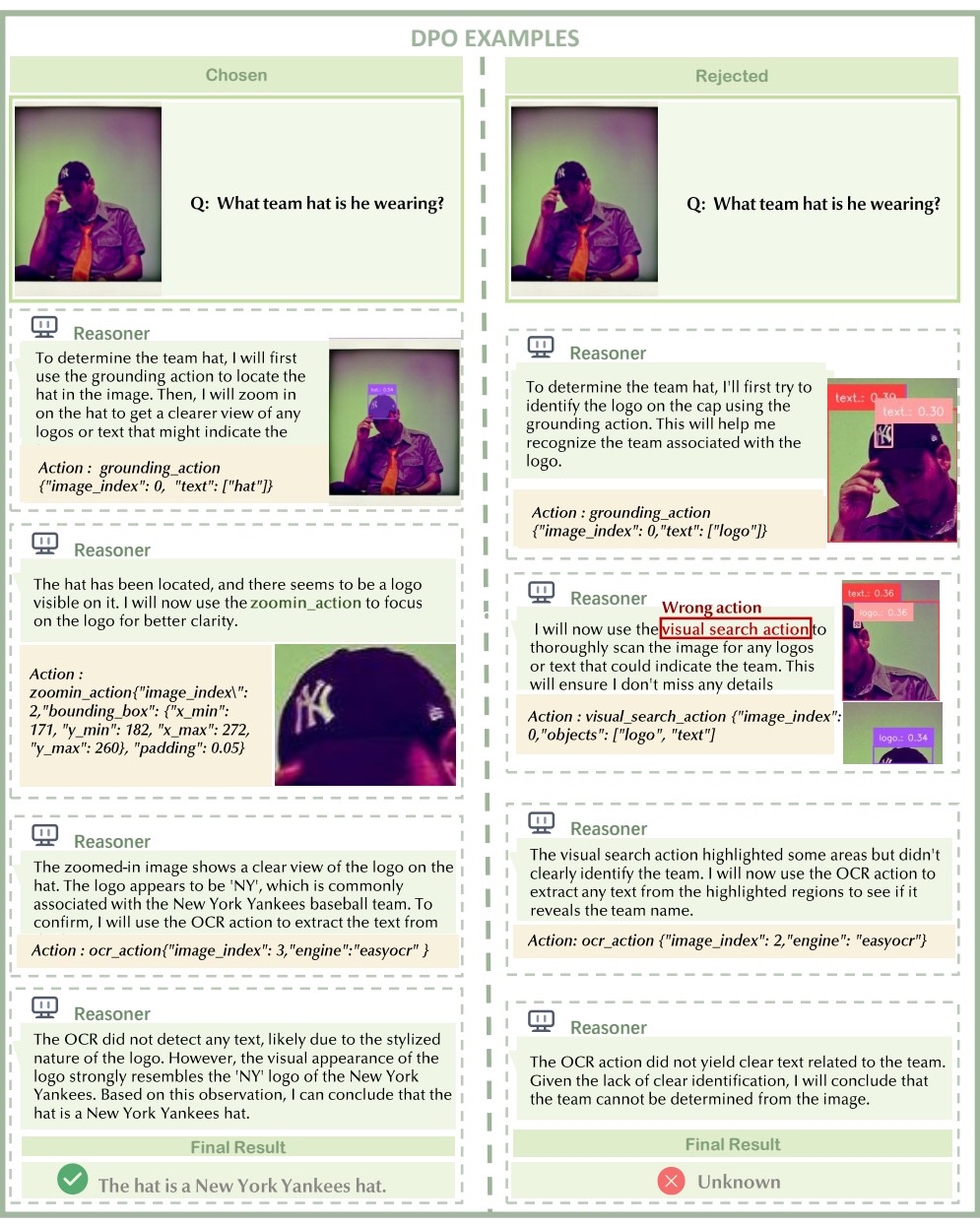

Figure 3: We present an example of the generated DPO data.

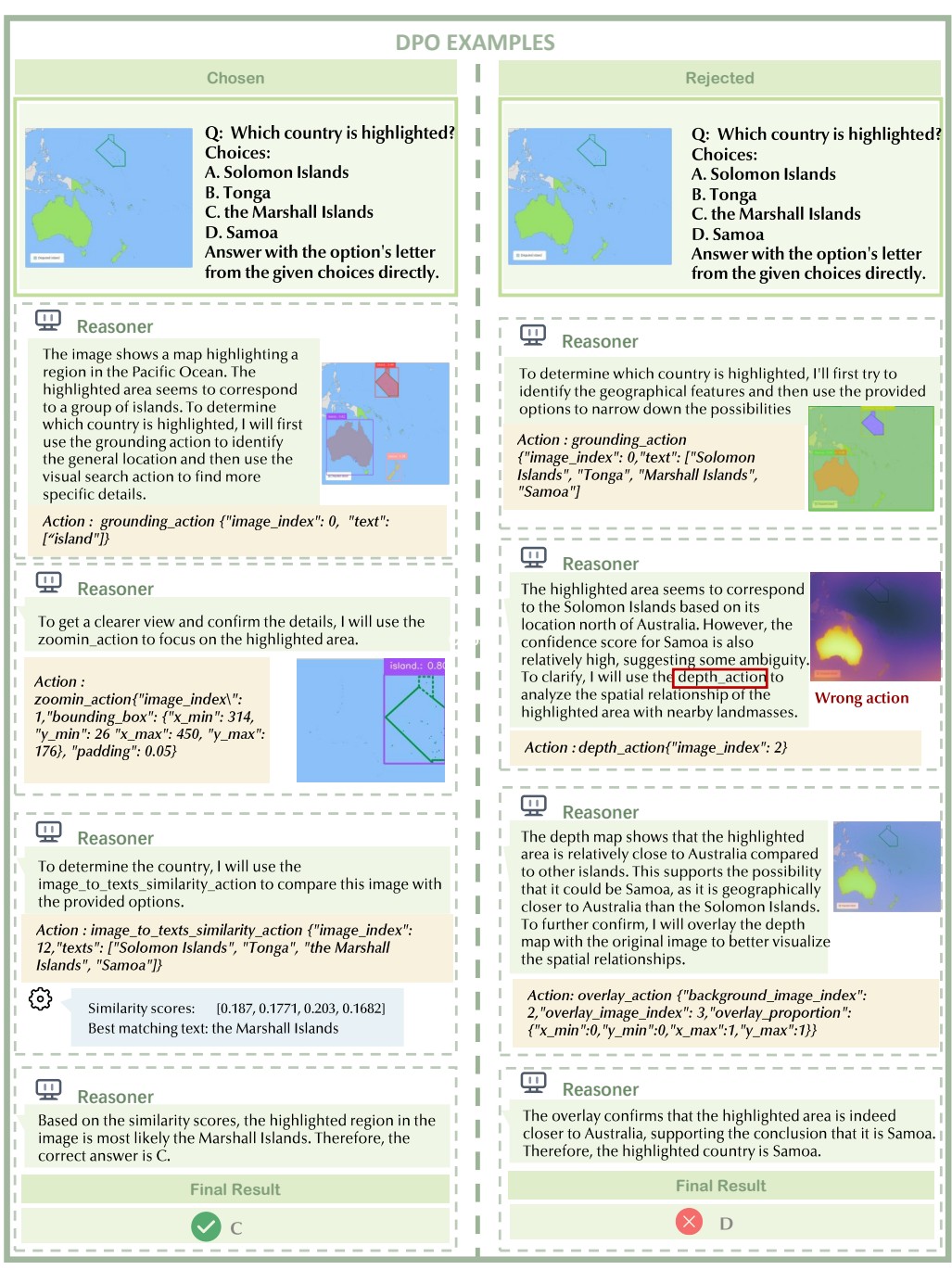

Figure 4: We present additional examples of the generated DPO data.

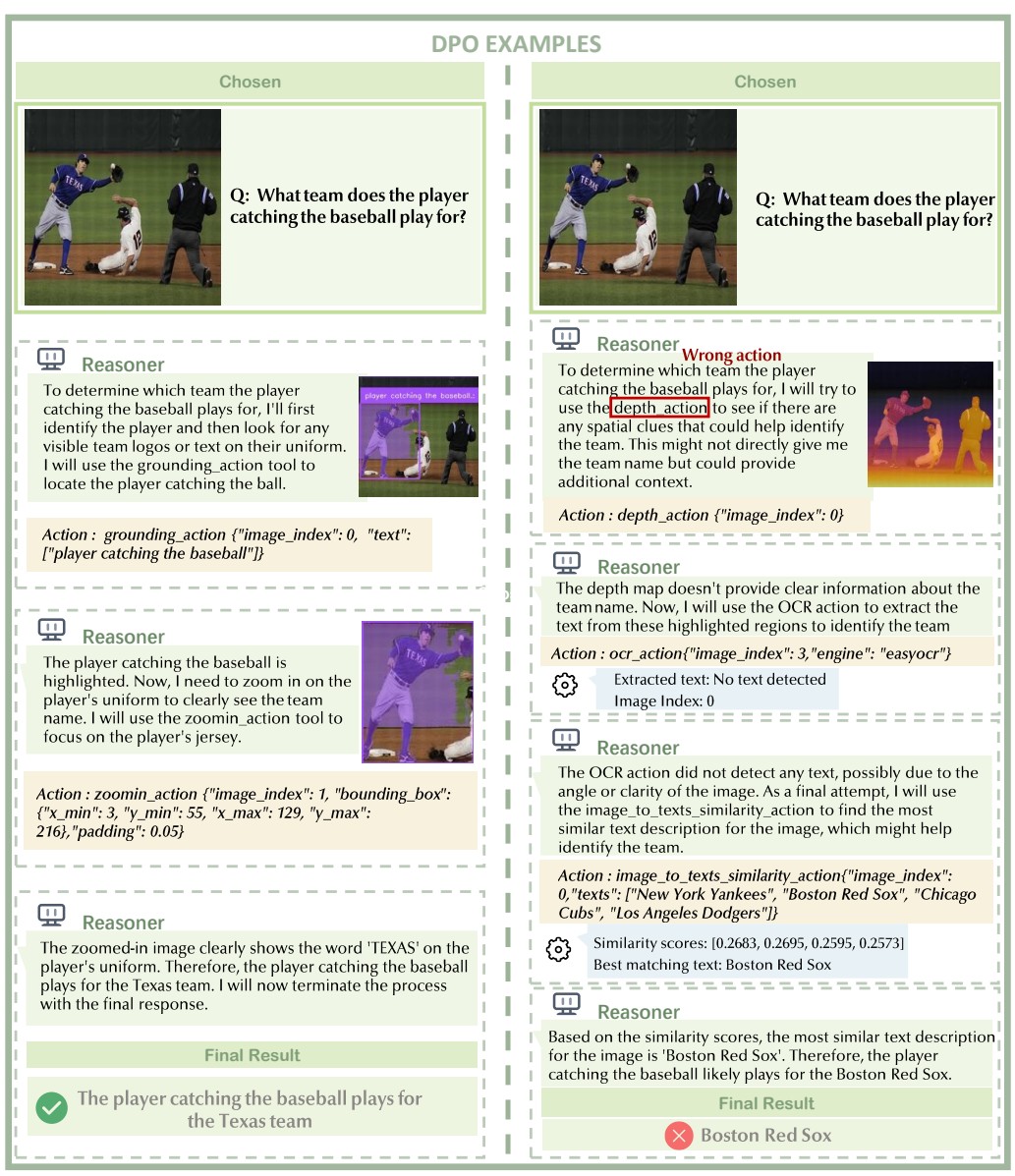

Figure 5: We present additional examples of the generated DPO data.

## C.2  VTS-SFT Dataset Examples

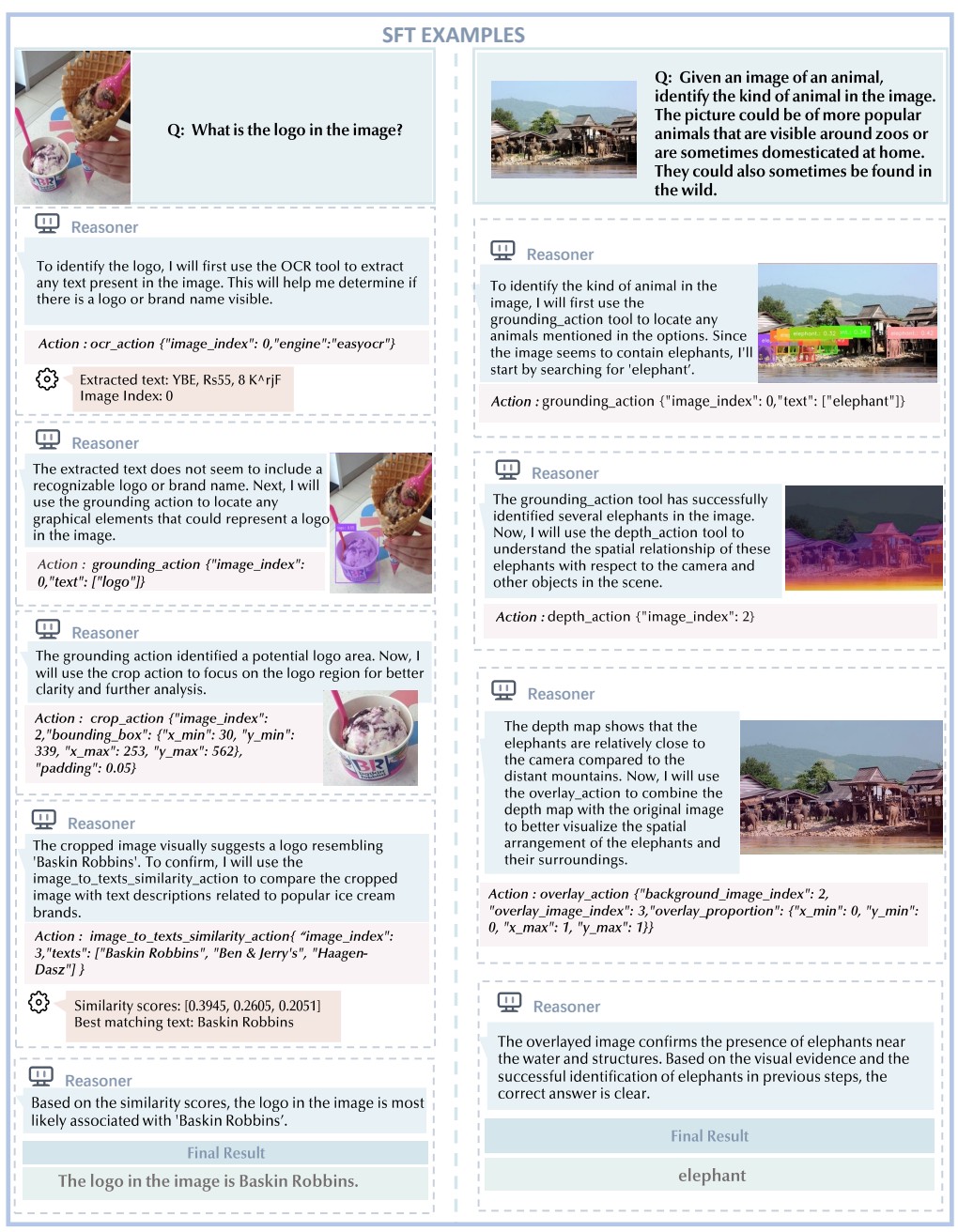

Figure 6: We present examples of the SFT data.

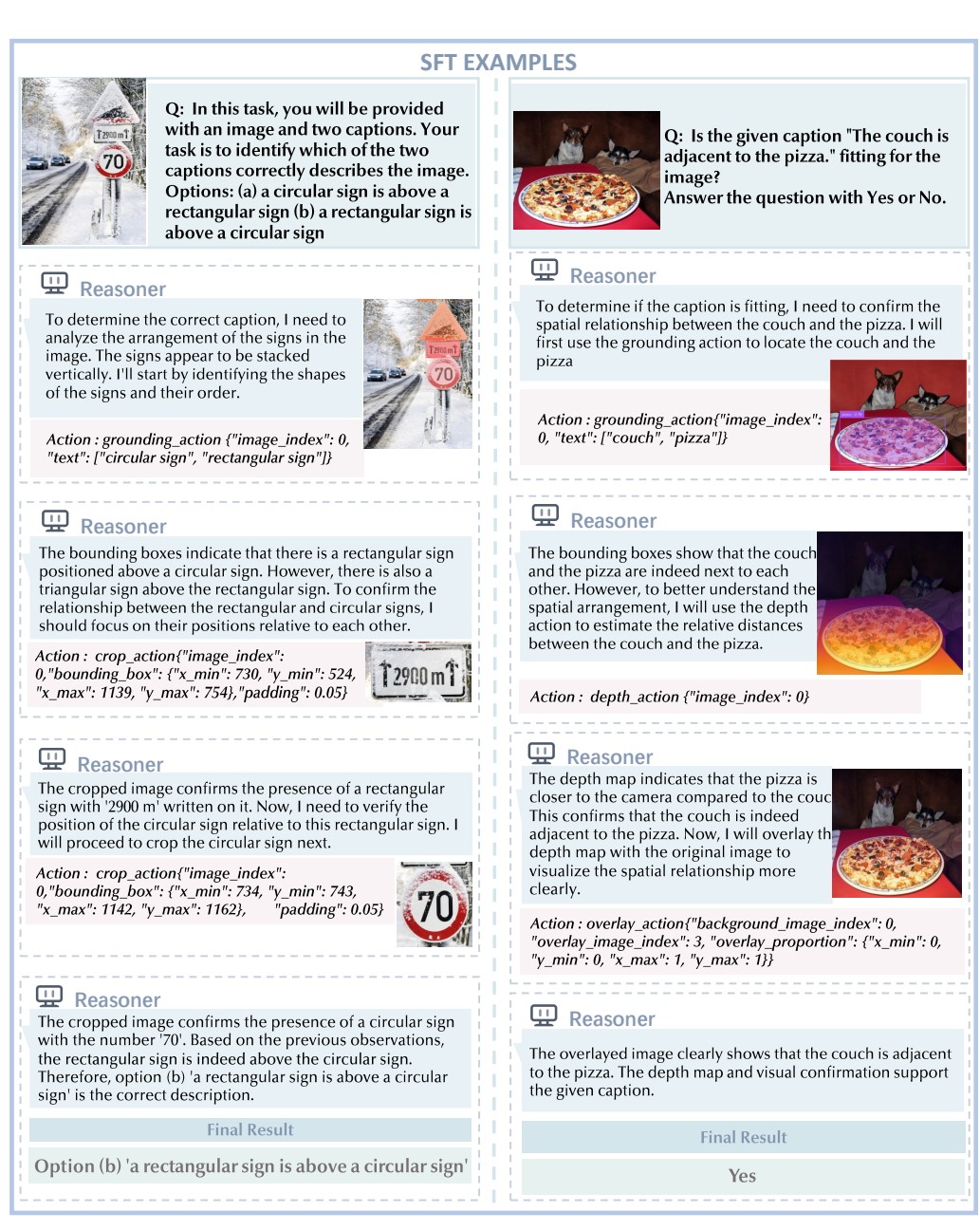

Figure 7: We present examples of the SFT data.

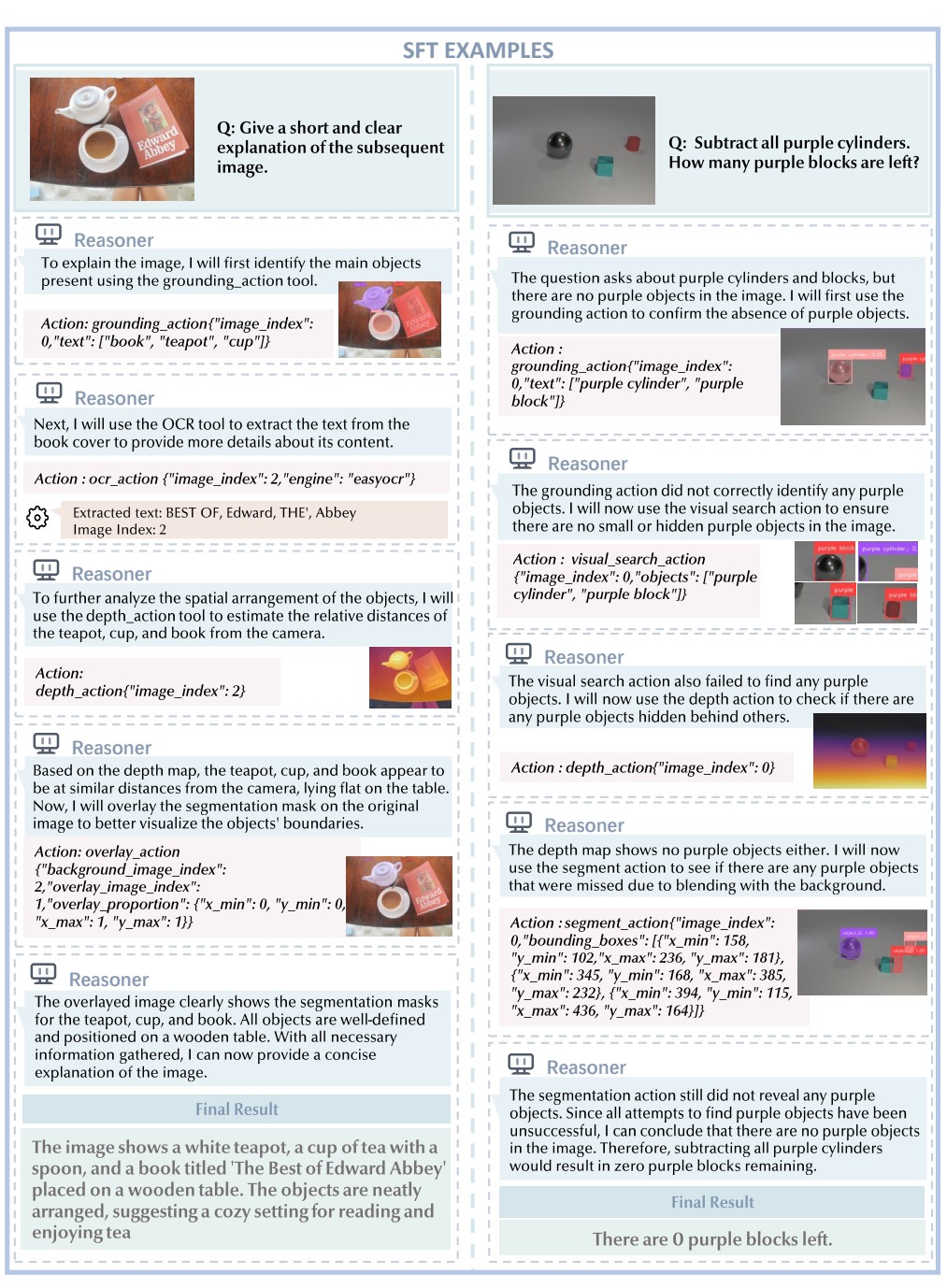

Figure 8: We present examples of the SFT data.

## C.3  Benchmark Examples

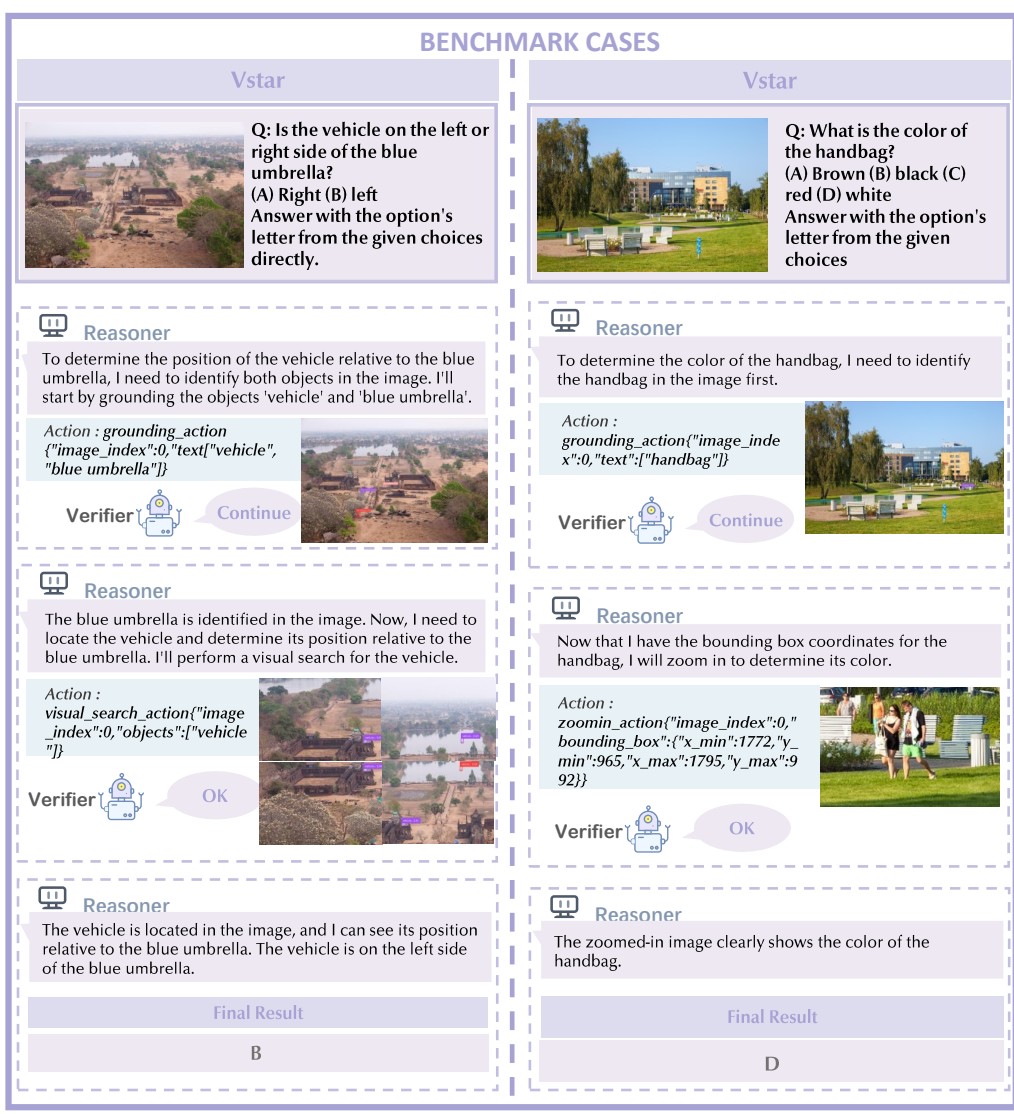

Figure 9: We demonstrate examples of reasoning on benchmark data using VTS-V.

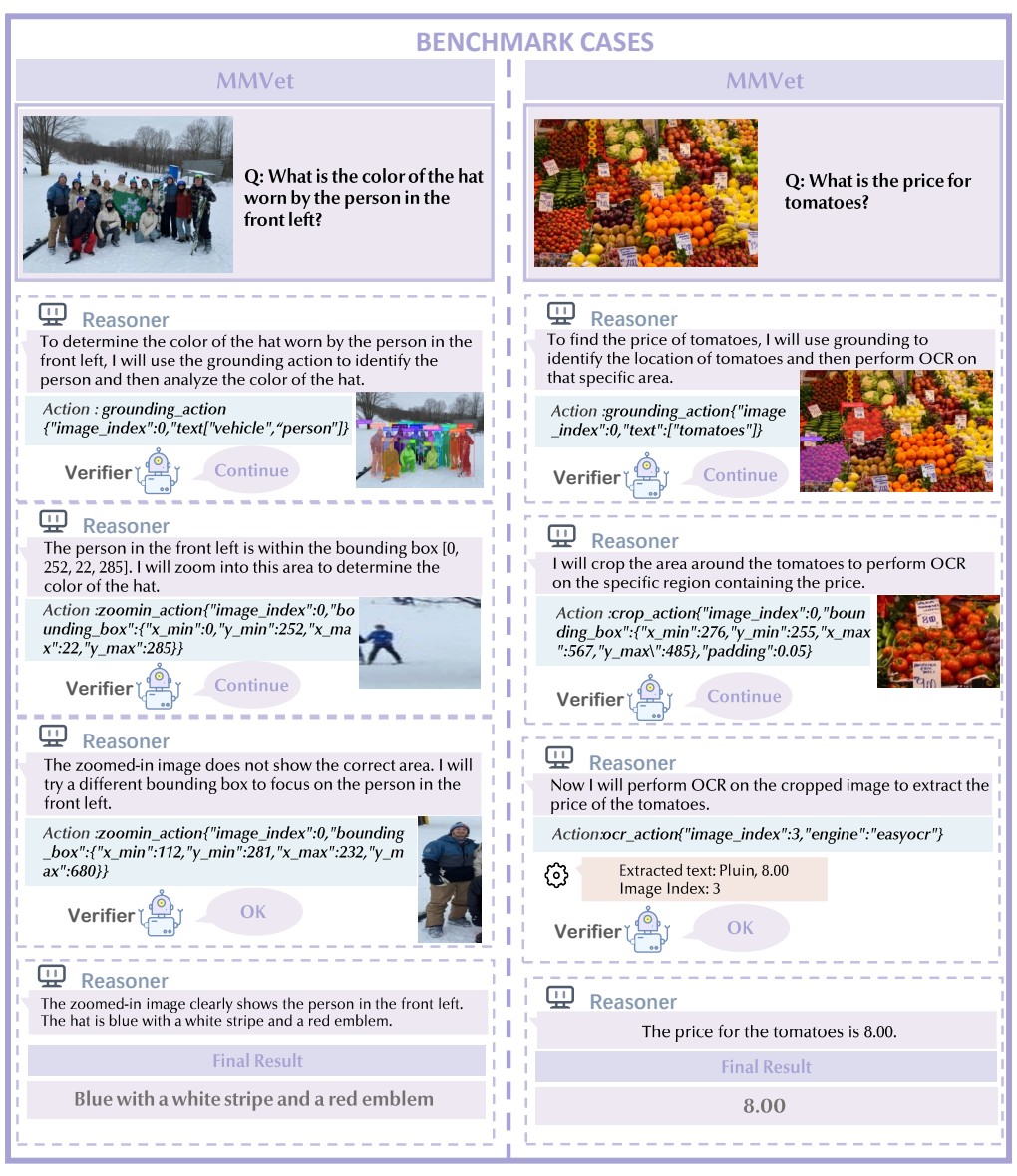

Figure 10: We demonstrate examples of reasoning on benchmark data using VTS-V.

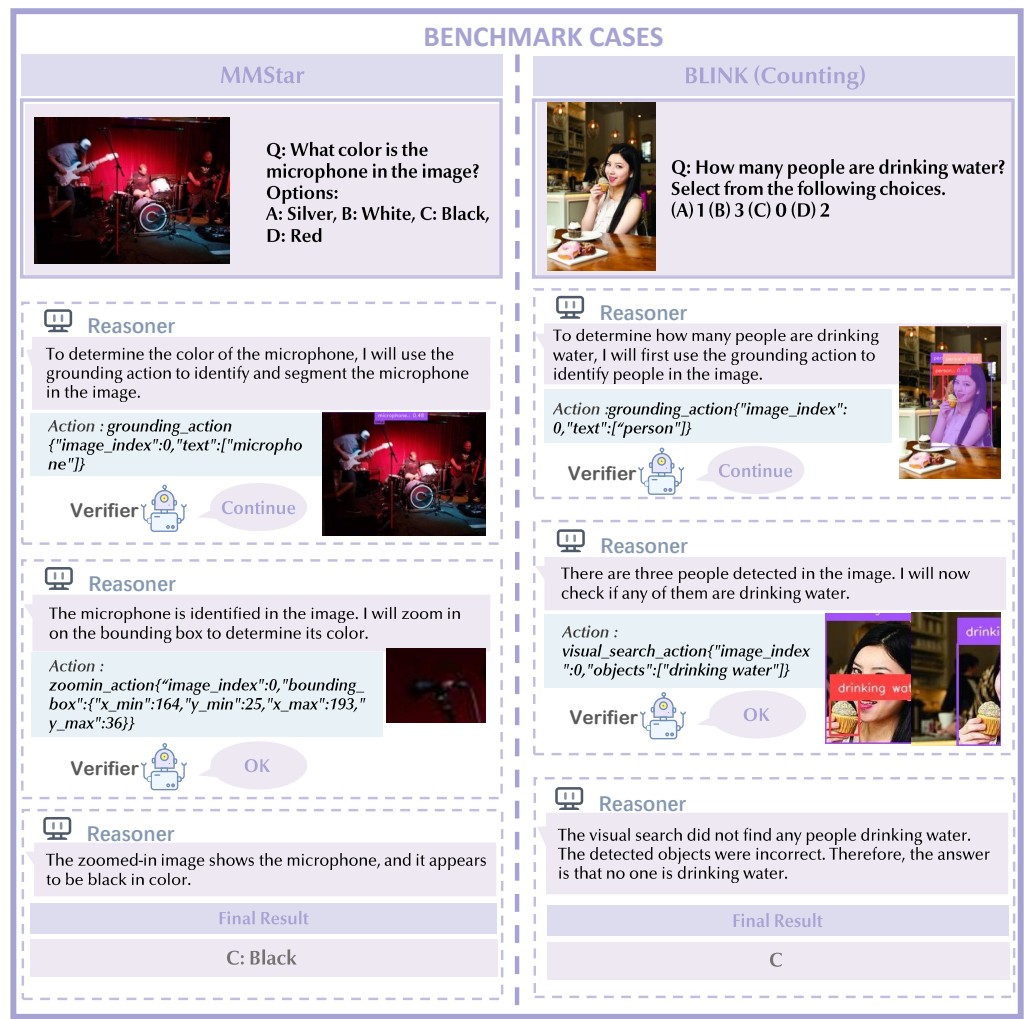

Figure 11: We demonstrate examples of reasoning on benchmark data using VTS-V.

## D  Broader Impact

Our work presents a foundational framework for enhancing visual reasoning via Visual Token Scaling with Verification (VTS-V). While not tied to a specific deployment, the technology could pose risks if misused. For instance, improved fine-grained image understanding may be applied in surveillance or disinformation pipelines, raising concerns about privacy, fairness, and potential for malicious use. To mitigate such risks, we recommend responsible release practices (e.g., usage restrictions, auditing tools, or gated access) and encourage further research on bias and misuse detection. Though the societal impact is limited at this stage, we acknowledge its importance as the technology evolves.

## E  Inference-Time Overhead Analysis

This section provides a detailed quantitative analysis of the inference-time overhead introduced by our VTS-V framework, including latency, memory usage, and the number of intermediate tokens generated during dynamic reasoning.

Our method employs iterative reasoning steps composed of two main components:

- **Verifier calls**: Require only prefill computation (forward pass over the input context to generate stop/continue decisions)

- **Reasoner calls**: Involve autoregressive decoding to produce tool instructions

The iterative interaction follows this structured process:

```
context = input_prompt: Q (image, text_question)
while verifier(context) == 'continue':
    current_output = reasoner(context)
    physical_feedback = tool_using_API(current_output)
```

We define the total inference latency as:

$$\text{Latency} = \sum_{i=1}^{H} \left( \frac{C_i}{P_t} + \frac{O_i}{D_t} \right) \tag{15}$$

Where:

- $H$: Total number of reasoning steps
- $C_i$: Number of input tokens at step $i$ (verifier prefill)
- $O_i$: Number of generated tokens at step $i$ (reasoner decode)
- $P_t$: Model's prefill throughput (tokens/sec)
- $D_t$: Model's decode throughput (tokens/sec)

For Qwen2.5-VL-7B-Instruct with VTS-V, we observe the following characteristics:

- Average steps ($H$): 3.1
- Average verifier prefill tokens: $\sim$10058.0 per step
- Average reasoner decode tokens: $\sim$95.3 per step
- Throughput on 80GB A800 GPUs (float16):
  - $P_t \approx 18493.6$ tokens/s
  - $D_t \approx 231.8$ tokens/s

The total latency is calculated as:

$$\text{Total Latency} \approx 3.1 \times \left( \frac{10058.0}{18493.6} + \frac{95.3}{231.8} \right) \approx 2.96 \text{ seconds}$$

Memory requirements for our framework include:

- With `max_model_len=65536` and `dtype=float16`, models can be deployed within 32GB through vLLM or SGLang on 80GB A800 GPUs
- All tool models (including GroundingDINO, Depth-Anything, etc.) collectively require approximately 23 GB of additional memory during inference

We observe a clear tradeoff between reasoning depth and computational cost, which is justified by significant accuracy gains (e.g., +7.58% average improvement on BLINK for Qwen2.5-VL-7B-Instruct). To mitigate latency for practical deployment, our framework supports several optimization strategies, such as parallel verifier batching, early stopping, and adaptive step truncation via $\epsilon$-threshold.

This analysis demonstrates that while our framework introduces additional computational overhead, the resulting accuracy improvements justify this cost, and practical deployment is feasible with appropriate optimizations.

# F Extended Evaluation on Broader Benchmarks

This section presents extended evaluation results of our VTS-V framework on additional benchmarks, providing a more comprehensive assessment of its generalization capabilities across diverse visual reasoning tasks.

## F.1 Experimental Setup

We conducted additional experiments evaluating GPT-4o augmented with our VTS-V framework on three challenging benchmarks: V*Bench [27], MMStar [3], and MathVista [18]. For comparison, we included the following baseline methods:

- **GPT-4o**: The base model without additional reasoning mechanisms
- **GPT-4o + CoT**: GPT-4o with Chain-of-Thought prompting [26]
- **GPT-4o + Sketchpad**: GPT-4o with Visual Sketchpad framework [11]

## F.2 Results and Analysis

Table 5 summarizes the performance comparison across different benchmarks:

Table 5: Performance comparison of GPT-4o variants on broader benchmarks. Best results are in **bold**.

| Method | V*Bench | MMStar | MathVista |
|---|---|---|---|
| GPT-4o | 60.73 | 58.40 | 63.80 |
| GPT-4o + CoT | 60.73 | 56.47 | **66.50** |
| GPT-4o + Sketchpad | 71.20 | 60.33 | 59.40 |
| GPT-4o + VTS-V (Ours) | **73.82** | **65.80** | 65.90 |

Key observations from these extended evaluations:

- Our VTS-V framework achieves state-of-the-art performance on both V*Bench and MMStar benchmarks, with absolute improvements of **13.09%** and **7.40%** over the baseline GPT-4o, respectively. These results demonstrate the effectiveness of our verifier-guided visual token scaling approach for complex visual reasoning tasks.
- On the MathVista benchmark, our method outperforms the base GPT-4o by **2.10%** and achieves competitive performance compared to GPT-4o + CoT, with only a **0.60%** difference from the top-performing method. This demonstrates the versatility of our approach across mathematical reasoning in visual contexts.
- The consistent improvements across diverse benchmarks highlight the generalization capability of our framework, which effectively enhances visual reasoning without task-specific adaptations.

# G Analysis of Performance Differences Between GPT-4o and Open-Source Models

This section analyzes the performance differences observed when applying VTS-V to GPT-4o versus open-source models without fine-tuning. While GPT-4o shows consistent improvements with VTS-V without additional training, open-source models like Qwen2-VL and LLaMA-3.2-Vision initially experience performance degradation. This discrepancy stems from several key factors:

- **Tool-Using Priors**: GPT-4o has extensive exposure to tool-augmented workflows during pretraining, enabling it to handle structured reasoning formats (e.g., Plan → Action → Tool call) naturally. Open-source models lack this prior and struggle to generate valid tool-use commands without fine-tuning.
- **Instruction Following Robustness**: GPT-4o's alignment tuning (e.g., RLHF) allows it to interpret multi-turn instructions and verifier signals effectively. Open-source models require fine-tuning to learn how to respond to verifier guidance correctly.

- **Empirical Recovery via Fine-Tuning**: After supervised fine-tuning on our VTS-SFT dataset, open-source models not only recover but exceed baseline performance (e.g., Qwen2VL-7B improves from 48.01% to 52.19%), demonstrating the dataset's role in enabling tool-use behaviors.

This analysis highlights the importance of our VTS-SFT dataset for adapting open-source models to verifier-guided reasoning, while confirming the framework's generality with state-of-the-art models like GPT-4o.

## H Visual Token Scaling Analysis with Optimal-Stopping Verifier

This section presents experimental results demonstrating the relationship between multi-step reasoning performance and visual token scaling, highlighting our framework's dynamic behavior and the effectiveness of the optimal-stopping verifier.

Table 6: Visual token scaling performance for GPT-4o on BLINK Counting task

| Visual & Text Tokens Per Sample | GPT-4o BLINK Counting Accuracy |
|---|---|
| 550 | 49.17% |
| 1100 | 58.33% |
| 1650 | **67.50%*** |
| 2200 | 64.16% |
| 2750 | 65.83% |

Table 7: Visual token scaling performance for open-source models on BLINK subtasks

| Visual & Text Tokens | Qwen2.5-VL Fun.Corr. | Qwen2-VL Depth | LLaMA-3.2 Spatial |
|---|---|---|---|
| 600 | 19.23% | 56.45% | 62.94% |
| 1200 | 23.85% | **60.48%*** | 65.03% |
| 1800 | 25.38% | 58.87% | **69.23%*** |
| 2400 | **30.00%*** | 57.26% | 67.83% |
| 3000 | 26.15% | 58.06% | 67.13% |

The tables illustrate the scaling relationship between reasoning accuracy and token usage across different models. On average, GPT-4o generates approximately 50 text tokens per reasoning step, while our trained open-source models generate around 100 text tokens per step, with both processing approximately 100 visual tokens through dynamic image operations per step.

Key observations from the scaling analysis:

- **Non-monotonic scaling**: Accuracy improves with increased visual tokens initially, but may decline with excessive reasoning steps, indicating diminishing returns.
- **Verifier-guided optimal stopping**: Our VTS-V verifier consistently identifies the best stopping points (marked by *), balancing reasoning depth and computational efficiency.
- **Robust performance**: Accuracy remains substantially higher than under-reasoning baselines even beyond optimal steps, demonstrating the benefits of sufficient visual grounding.

These results validate our verifier's ability to dynamically scale visual tokens while maintaining optimal reasoning performance across different model architectures and task types.

## I Robustness and Generalization Analysis of the Verifier

This section provides additional analysis of the verifier's robustness and generalization capabilities, addressing its performance on unseen domains and tasks beyond the training distribution.

The VTS-DPO dataset is constructed from the comprehensive LLaVA-OneVision (LLaVA-OV) dataset, which spans over 80 distinct vision-language task types. Our curated DPO dataset comprises 301,028 preference pairs that cover a diverse range of visual reasoning domains, ensuring broad coverage and representation.

Table 8: VTS-DPO dataset composition across different domains

| Domain Category | # Tasks | # Examples |
|---|---|---|
| General VQA | 25 | 135,804 |
| Document/Chart/Screen | 18 | 100,681 |
| Math/Logical Reasoning | 15 | 52,049 |
| OCR/Text-grounded Tasks | 6 | 12,494 |

The dataset encompasses diverse visual reasoning capabilities including document understanding, scientific diagram interpretation, mathematical problem solving, chart question answering, optical character recognition, and general commonsense visual question answering.

Importantly, all evaluation benchmarks used in our experiments (BLINK, V*Bench, MMStar, and MathVista) are held-out from the training data, ensuring that the verifier is tested on completely unseen reasoning tasks and domains. The consistent performance improvements observed across these diverse benchmarks demonstrate the verifier's strong generalization capabilities beyond its training distribution.

The comprehensive coverage of VTS-DPO, combined with the held-out evaluation protocol, validates the robustness of our verifier training approach and its applicability to novel visual reasoning scenarios.

## J    Separate Reasoner and Verifier

This section explains our design decision to maintain separate reasoner and verifier components rather than employing a single unified model. While a unified approach offers potential efficiency benefits, our analysis demonstrates that a decoupled architecture provides superior performance for multi-modal, tool-augmented reasoning tasks.

Reasoning and verification represent fundamentally distinct capabilities: the reasoner must generate executable tool-using action plans conditioned on complex multi-modal contexts, while the verifier must evaluate the overall quality and utility of reasoning steps. Empirical observations show that when both responsibilities are assigned to a single model, performance typically deteriorates as the model struggles to maintain both generative planning and discriminative evaluation capabilities.

Experimental results provide strong validation for the separated architecture. As shown in Table 9, adding a standalone verifier module to GPT-4o yields substantial performance gains:

Table 9: Performance comparison of GPT-4o with and without separate verifier on BLINK benchmark

| Model | Avg Accuracy (BLINK) |
|---|---|
| GPT-4o | 62.25% |
| GPT-4o + Verifier | **69.15%** (+6.90%) |

These improvements are particularly pronounced on tasks requiring structured multi-step reasoning, with gains of +18.33% on Counting and +12.91% on Functionally-Correlated tasks. Beyond performance benefits, the separated architecture offers practical advantages including model-agnostic verifier deployment across architectures (GPT-4o, Qwen, LLaMA), reduced fine-tuning costs, and enhanced extensibility for new tasks and domains. This analysis confirms that the separated reasoner-verifier design provides both performance benefits and practical advantages for complex visual reasoning tasks.

## K    Interpretability of Reasoning Traces

This section clarifies our claim regarding the improved interpretability of reasoning traces generated by our VTS-V framework. We define interpretability in this context as the explicit, grounded, and stepwise nature of the reasoning process that makes the model's decision-making transparent and verifiable.

Traditional vision-language models perform latent, one-shot image encoding and generate answers based on static visual token embeddings, offering limited insight into their reasoning process. In contrast, our framework produces multi-turn reasoning traces composed of explicit tool-use steps, where each turn includes:

- A verbalized subgoal or planning statement

- A specific tool invocation with parameters

- Visual feedback or observation from tool execution

These intermediate outputs are grounded in actual visual content through operations such as region cropping, OCR text extraction, depth map generation, and spatial analysis. This structured output format provides a transparent reasoning trajectory that allows both human users and automated systems to inspect the decision process at each step.

Empirical evidence supports this improved interpretability. Across multiple benchmarks, our models generate longer and more structured reasoning trajectories compared to vanilla baselines. For instance, the average reasoning steps increase from approximately 1.5 (GPT-4o baseline) to 3.2 (our VTS-V framework), with diverse tool usage patterns including grounding actions, depth analysis, OCR operations, and region cropping.

Examples of these interpretable reasoning traces are visible in Figure 1 and Appendix C.3 of the main paper, demonstrating how models produce dynamic reasoning involving tool selection and verification-guided iteration. While future work could include manual evaluations of reasoning quality, our current framework already provides verifiable, human-readable sub-decisions that represent a significant advancement beyond traditional black-box VLM inference.

## L    Ablation Study on the Reasoner Module

This section presents an ablation study evaluating the specific contribution of the reasoner module within our VTS-V framework. We compare two configurations: models using verifier guidance without the trained reasoner (tool actions generated directly by the base model) versus our full VTS-V framework with both trained reasoner and verifier components.

Table 10: Performance comparison with and without the reasoner module

| Model Variant | V*Bench | MMStar | MathVista |
|---|---|---|---|
| Qwen2.5VL-7B + Verifier (w/o Reasoner) | 67.54 | 55.93 | 7.80 |
| Qwen2.5VL-7B + VTS-V (w/ Reasoner) | **75.13** | **57.93** | **23.52** |
| Qwen2VL-7B + Verifier (w/o Reasoner) | 50.80 | 44.29 | 20.24 |
| Qwen2VL-7B + VTS-V (w/ Reasoner) | **66.67** | **55.26** | **23.80** |

The results demonstrate that incorporating the trained reasoner significantly improves performance across all benchmarks. Without the reasoner module, models struggle to perform reliable multi-step reasoning, particularly on complex tasks like MathVista where performance degrades substantially.

The addition of the reasoner consistently produces large performance gains, with improvements of up to +15.72 points on MathVista for Qwen2.5VL-7B and +15.87 points on V*Bench for Qwen2VL-7B. These findings highlight the critical importance of the reasoner module in executing valid tool-use actions and effectively following verifier guidance during multi-step reasoning processes.

## M    Comparison with Dynamic Visual Attention Methods

This section provides a comparative analysis between our VTS-V framework and prior work on dynamic visual attention, highlighting key distinctions in methodology and capabilities.

Our framework introduces a general, verifier-guided inference-time reasoning process based on a Markov Decision Process (MDP) formulation, enabling multi-step refinement of visual understanding through a flexible set of visual tools. This approach differs fundamentally from existing methods in several important aspects:

- **LMEye**: While LMEye allows large language models to request new visual evidence, it lacks a structured action space and formal termination control. Our approach models perception as MDP-driven tool selection with a multi-step DPO-trained verifier that dynamically determines when to terminate reasoning.

- **InstructBLIP**: This method enhances initial visual encoding through instruction tuning but performs single-pass perception. In contrast, our framework supports iterative, adaptive reasoning during inference, allowing models to invoke diverse tools (OCR, zoom, grounding) based on evolving contextual understanding.

- **V\***: Primarily designed for recursive visual search to locate objects in high-resolution images, V\* operates with a binary search objective. Our framework addresses a broader range of tasks including mathematical reasoning, OCR, and chart understanding, with termination based on verifier-guided reasoning quality rather than search completion.

Our core innovations distinguish VTS-V from these approaches:

1. **Inference-time visual token scaling**: Dynamically adjusts which image regions receive detailed processing through tool invocation, rather than using fixed input-time processing.

2. **MDP-based multi-step reasoning**: Formally defines each state-action step to support structured perception and tool-based decision-making with theoretical guarantees.

3. **Verifier-guided termination**: Employs a learned verifier trained via multi-step DPO to provide step-wise feedback and enforce bounded reasoning (Lemma 3.2), improving both efficiency and robustness.

These innovations position VTS-V as a more comprehensive and theoretically grounded approach to dynamic visual reasoning compared to existing methods.

## N  Novelty Analysis and Comparison with Prior Work

In this section, we provide a detailed analysis of the novelty of our VTS-V framework and its distinctions from related works. While our method shares some conceptual similarities with prior research in iterative reasoning, reasoner-verifier separation, and tool use, it introduces several key innovations that advance the state of the art in multi-step visual reasoning:

- **MDP-based Reasoning Framework for Dynamic Visual Token Scaling**: Unlike V\* [27] which primarily focuses on task-specific visual cropping, we formulate multi-step visual reasoning as a *Markov Decision Process* over a diverse and extensible set of visual tools (e.g., OCR, grounding, depth, segmentation). This formulation enables dynamic visual token scaling—allowing the model to adaptively focus on different regions or modalities at each reasoning step—across a wide range of tasks and model architectures.

- **Step-wise Verifier with Theoretical Guarantees**: While V-STaR [9] employs a verifier to rank final answers, our verifier operates **at each reasoning step**, providing real-time guidance and enabling early termination with theoretically bounded steps (Theorem 3.2). This ensures both interpretability and computational efficiency, a feature not present in prior verifier-based methods.

- **First Dataset for Multi-Step Visual Tool Reasoning**: Existing tool-use datasets (e.g., from Visual Program Distillation [11]) do not support multi-turn visual reasoning with rich intermediate states. We introduce VTS-SFT and VTS-DPO, the first datasets designed for fine-grained tool selection and verifier training over more than 80 diverse vision-language tasks, enabling robust generalization and grounding.

- **Generalization Without Fine-Tuning**: Our framework generalizes effectively even to closed-source models like GPT-4o without additional fine-tuning, improving performance by **+6.9%** on the BLINK benchmark. This demonstrates the flexibility and model-agnostic nature of our approach.

In summary, while our work builds upon prior art, it integrates these ideas into a unified, theoretically sound, and empirically validated framework that supports multi-step, tool-augmented, verifier-guided visual reasoning. We believe this represents a meaningful step forward in enabling fine-grained and interpretable reasoning in multimodal systems.

