# OpenReview forum: "Multi-step Visual Reasoning with Visual Tokens Scaling and Verification"
_NeurIPS.cc/2025/Conference — NeurIPS 2025 poster_

### Official Review · Reviewer_CE1C · 2025-07-01

**Clarity:** 2
**Significance:** 2
**Originality:** 2
**Rating:** 4
**Confidence:** 4

**Summary:**

This research addresses the limitations of the static visual encoding paradigm in current Multimodal Large Language Models (MLLMs). It highlights how their pre-processing approach, which relies on fixed visual tokens, hinders dynamic reasoning capabilities and fundamentally differs from the iterative feedback mechanism of human perception. The authors propose a framework: inference-time visual token scaling. This framework models the process using a Markov Decision Process and dynamically adjusts the inference process with a multi-step DPO-trained validator. This allows the model to achieve adaptive, refined analysis of visual content. To validate the effectiveness of their method, the researchers also constructed a specialized dataset called VTS, which includes both supervised trajectories and preference comparisons.

**Questions:**

see the weakness

**Ethical Concerns:**

["NO or VERY MINOR ethics concerns only"]

**Final Justification:**

My concerns have now been addressed. The authors promise to revise the paper further to include some new complementary experiments and discussions with other methods.

**Paper Formatting Concerns:**

N/A, figure 1 and 2 should be more clear.

**Quality:**

2

**Strengths And Weaknesses:**

**Strengths**

1. The paper proposes a dataset which may help the training of the reward model.

2. The proposed method is effective for some image-based visual reasoning tasks


**Weakness**

1. Figures 1 and 2 are not clear.

2. If the "reasoning reinforcement polishing" (likely referring to your iterative refinement process) involves multiple rounds of actions, please elaborate on the key distinctions between your proposed reinforcement learning method and existing multi-round reinforcement learning approaches. Specifically, how does your inference-time visual token scaling mechanism and the use of a multi-step DPO trained validator differentiate it from, or build upon, traditional multi-round reinforcement learning paradigms?

3. Your core motivation is to address the limitations of static visual information input, which leads to incomplete attention during contextual reasoning. While the self-attention mechanism in large language models does allow for varying attention to visual tokens during training and inference, your work introduces a novel approach to dynamic attention adjustment for visual information. It's crucial to acknowledge and explicitly compare your method with prior work that has also tackled dynamic visual attention or iterative perception in MLLMs. Please discuss how your approach, particularly inference-time visual token scaling and the Markov Decision Process modeling, differs from and potentially advances solutions proposed in works such as:
LMEye: Iterative Perception Network; InstructBLIP; V*
A clear comparative analysis against these established methods would significantly strengthen your paper's contribution and highlight the novelty of your proposed framework.

4. The experimental results show limited improvement in some datasets, and the improvement may be largely relevant to the sft stage.

---

> ### Author Rebuttal · Authors · 2025-07-31
>
> **Overall Response**
>
> We appreciate the insightful feedback and detailed questions. In response, we clarified key conceptual differences from traditional RL and dynamic attention methods, emphasizing our verifier-guided, multi-step MDP formulation and inference-time visual token scaling. We also updated Figures 1 and 2 for clarity, and clarified the respective roles of our framework and fine-tuning, supported by both GPT-4o and open-source model results. These clarifications and improvements will be included in the revised version.
>
> > **Response to W1: Clarity of Figures 1 and 2**
> >
>
> Thanks for the feedback. We have improved the clarity of Figures 1 and 2, and the revised versions will appear in the updated paper.
>
> > **Response to W2: Differences from Traditional Multi-Round Reinforcement Learning**
> >
>
> Thanks for the thoughtful question. While our method is formally cast as a Markov Decision Process (MDP), it departs significantly from traditional multi-round reinforcement learning (RL) paradigms in both its state formulation and guidance mechanism.
>
> **1. Dynamic Visual Token Scaling at Inference Time**: Traditional RL agents operate over fixed state encodings (e.g., static image tokens). In contrast, our framework performs inference-time visual token scaling, allowing the model to dynamically generate new visual tokens through actions such as zooming, OCR, grounding, and depth estimation.
>
> - This enables iterative refinement of visual context, rather than reasoning over a one-shot encoding.
> - Actions are selected conditionally based on prior steps, supporting context-aware visual exploration.
>
> **2. Verifier-Guided Reasoning via Multi-Step DPO (Not Reward Maximization)**: Instead of using scalar rewards to optimize policy, we guide reasoning using a multi-step DPO-trained verifier, which evaluates entire reasoning trajectories based on preference comparisons.
>
> - The verifier provides step-wise utility signals, assessing both intermediate tool actions and final outputs.
> - This enables fine-grained control and inference-time termination based on theoretical bounds (Theorem 3.2), which is typically lacking in RL agents prone to over-exploration or loops.
>
> We will clarify these distinctions in the revised draft.
>
> > **Response to W3: Comparison with Prior Work on Dynamic Visual Attention**
> >
>
> Thanks for raising this important point. We agree that dynamic visual perception is a growing area, and we appreciate the opportunity to clarify how our framework differs from prior methods like LMEye, InstructBLIP, and V*.
>
> Our proposed framework introduces a general, verifier-guided inference-time reasoning process built on a Markov Decision Process (MDP). It enables multi-step refinement of visual understanding via a flexible set of visual tools—something that existing approaches do not fully support.
>
> **Comparison with Prior Work:**
>
> - LMEye allows LLMs to request new visual evidence, but lacks a structured action space and termination control. In contrast, we model perception as MDP-driven tool selection and pair it with a multi-step DPO-trained verifier that decides when to stop.
> - InstructBLIP enhances initial visual encoding using instruction tuning, but performs single-pass perception. Our approach allows iterative, adaptive reasoning during inference, where the model can invoke tools like OCR, zoom, or grounding based on ongoing context.
> - V* performs recursive visual search, mainly for locating objects in high-res images. Our framework is more general-purpose, supporting a wider range of tasks (e.g., math, OCR, charts), and termination is based on verifier-guided reasoning quality, not a binary search objective.
>
> **Our Core Innovations:**
>
> 1. **Inference-Time Visual Token Scaling** – Dynamically adjusts which image regions are processed in more detail via tool-use, not fixed at input time.
> 2. **MDP-Based Multi-Step Reasoning** – Each state-action step is formally defined, supporting structured perception and tool-based decision-making.
> 3. **Verifier-Guided Termination** – A learned verifier trained via multi-step DPO provides step-wise feedback and enforces bounded reasoning (Theorem 3.2), improving efficiency and robustness.
>
> We will update the manuscript to explicitly include this comparative discussion.
>
> > **Response to W4: Clarifying Dataset-Wise Improvements and the Role of SFT**
> >
>
> Thanks for pointing this out. We agree that some datasets show more modest improvements, and we welcome the opportunity to clarify the role of our framework vs. fine-tuning in these results.
>
> As shown in Table 1, applying the VTS-V framework to GPT-4o—without any fine-tuning—already results in substantial gains:
>
> | **Model** | **Avg Accuracy (BLINK)** |
> | --- | --- |
> | GPT-4o | 62.25% |
> | GPT-4o + VTS-V | **69.15% (+6.9)** |
>
> This demonstrates that the framework itself is effective, particularly when the base model (like GPT-4o) is well-aligned and capable of interpreting multi-step instructions and tool-use signals.
>
> In Table 2, open-source models (e.g., Qwen2-VL, LLaMA3.2) show degraded performance when using VTS-V without fine-tuning. This is not due to the framework’s limitations, but rather:
>
> - These models lack tool-use priors and have no exposure to structured reasoning loops (e.g., Plan → Action → Tool → Feedback → Stop/Continue).
> - They also lack the instruction-following robustness of GPT-4o and thus cannot properly interpret verifier feedback or generate well-formed tool commands.
>
> Once fine-tuned with our VTS-SFT dataset, these models learn to align with verifier signals, producing valid reasoning trajectories and improving beyond their baselines (e.g., Qwen2VL-7B improves from 48.01% to 52.19%).
>
> We will clarify this distinction in the revision.

---

> > ### Comment · Reviewer_CE1C · 2025-08-05
> > **Respond to Authors**
> >
> > Thank you to the Authors for your rebuttal; my concerns have now largely been addressed. I'll improve my score to 4.

---

> > > ### Author Response · Authors · 2025-08-05
> > > **Thanks for your reply!**
> > >
> > > Thank you so much for your thoughtful feedback and for raising your rating. We sincerely appreciate your time and effort in evaluating our work, and we’re grateful that our rebuttal addressed your concerns. We will diligently implement your suggestions to refine the manuscript for its final version.

---

> ### Comment · Area_Chair_NFXJ · 2025-08-03
>
> Dear Reviewer,
>
> Could you please check if the authors’ rebuttal adequately addresses your concerns? If so, kindly acknowledge the rebuttal and provide any additional comments. If not, it would be greatly appreciated it if you could engage in a discussion with the authors. Your input at this stage is essential to the review process. Thank you very much for your time and effort!
>
> AC

---

> ### Author Response · Authors · 2025-08-05
> **Kind Reminder: Your Additional Thoughts on Paper #9056 Are Welcome**
>
> Dear Reviewer  CE1C,
>
> Your expert feedback is crucial to refining this work. While we fully understand the discussion period may pose challenges for your schedule, we would value the chance to clarify any final points with you prior to its conclusion on Aug 8.
>
> We hope we've been able to address your questions and concerns so far. We would be glad to address any further concerns you may have, and we will try our best to clarify promptly.
>
> Thank you again for your feedback and comments; they were really helpful!
>
> Warm Regards, Authors of Submission #9056

---

> ### Comment · Area_Chair_NFXJ · 2025-08-06
>
> Dear Reviewer,
>
> Please remember to submitting the *Mandatory Acknowledgement*. Thank you!
>
> AC

---

### Official Review · Reviewer_XMPf · 2025-07-01

**Clarity:** 3
**Significance:** 3
**Originality:** 2
**Rating:** 4
**Confidence:** 4

**Summary:**

This paper focus on inference-time visual token scaling, via a tool-augmented VLM Reasoner module and a verifier optimized by DPO. The Reasoner can be either API-based models or finetuned open-source models.
The proposed framework relies on the self-constructed dataset, a new tool-enhanced CoT-style dataset.
This paper also describes engineering details of their data generation pipeline.
The method outperforms baselines, both tool-augmented and prompt-based methods, across several visual reasoning benchmarks, demonstrating improved accuracy and tool-using ability. The authors also provide theoretical guarantees that their algorithm ensures a bounded number of reasoning steps, preventing looping or repetitive outputs.

**Questions:**

1. Please response to Weaknesses.
2. From Table2, there could be a conclusion that the improvement main comes from the curated dataset. The gain of the framework itself remains unclear.
3. Is it possible to add additional experiment of w/o reasoner (as the other ablation study in table3)? This could further demonstrate the effectiveness of the entire framework, otherwise the reasoner could be a dispensable module.

I acknowledge that the dataset could be useful, please provide further explanation and evidence to demonstrate the necessity and effectiveness of the framework (as mentioned above) , so that this paper would not remain significant in dataset engineering only. I will raiss my rating if all questions and concerns are solved.

Additional suggestion: (NOT weakness)
1. Some simple visualisations of the framework would assist in understanding of the entire work.
2. Please carefully check the tables and figure in appendix. They are not properly displayed.

**Ethical Concerns:**

["NO or VERY MINOR ethics concerns only"]

**Final Justification:**

The authors provide strong evidence supporting the effectiveness of the Reasoner and their proposed framework, which are my primary concerns. I will still increase my rating from 3 to 4 because authors provide persuasive technical clarification for most of my concerns.

**Limitations:**

Yes

**Paper Formatting Concerns:**

1. Tables in Appendix D exceed the colume width, and are not completely displayed.

**Quality:**

2

**Strengths And Weaknesses:**

**Strengths:**
1. The paper presents a technically solid framework with clear theoretical explanation.
2. The design and construction of tool-augmented multi-step visual reasoning dataset is well motivated, addressing a notable gap in existing datasets, which would be useful to future research.
3. The proposed method largely improves the performance of open-source models regarding visual reasoning and grounding tasks.

**Weaknesses:**
1.Authors claim that their proposed method offers more interpretable reasoning traces. However, no related evaluation or demonstrations of resulting generations of the trained models are provided (authors only provide the example trace in their constructed data).
  - More discussion about the results from the trained models using their data should be provided to better support the claim of "more interpretable reasoning traces". A dataset with longer reasoning traces or traces with diverse tools does not guarantee that the trained models would exhibit exactly desired performance.
  - Manual evaluation or further comparison is required to support the contribution regarding the term, "more interpretable reasoning trace". For example, in Line 92-95, these methods is not **limited** to 1-2 steps, many questions can be well solved only via 1-2 steps.
2. The verifier is similar to existing feedback-based reflective mechanisms. According to Line 103, the proposed method should differ in both tasks and training datasets. However, the proposed framework is also a tailored dataset whose actions are restricted to tool-use.

---

> ### Author Rebuttal · Authors · 2025-07-31
>
> **Overall Response**
>
> We appreciate the thoughtful feedback and valuable suggestions. In response, we clarified our definition of interpretability, highlighted how our structured tool-use traces offer stepwise, visually grounded transparency, and provided empirical evidence supporting this claim. We also clarified how our verifier differs from prior reflective mechanisms, distinguished the impact of the VTS-V framework from the dataset, and added ablations isolating the reasoner’s contribution. All corresponding updates and clarifications will be incorporated into the final version if the paper is accepted.
>
> > **Response to W1: Evaluation of the Claim “More Interpretable Reasoning Traces”**
> >
>
> We thank the reviewer for raising this important question regarding the interpretability of reasoning traces generated by our model. We clarify below what we mean by *interpretability* in the context of our method and why it is supported by the proposed framework and results.
>
> **1. Nature of Interpretability in Our Setting:** Traditional VLMs perform latent, one-shot image encoding and generate answers based on static visual token embeddings. In contrast, our model produces *multi-turn reasoning traces* composed of explicit tool-use steps:
>
> - Each turn consists of a verbalized subgoal, an invoked tool, and a visual feedback observation.
> - These intermediate outputs are grounded in the actual visual content (e.g., cropped regions, OCR text, depth maps), providing a transparent reasoning trajectory.
>
> This tool-augmented, structured output allows both humans and verifiers to inspect the decision process at each step. As a result, the model’s internal reasoning becomes observable and interpretable, rather than remaining a latent process.
>
> **2. Examples Beyond Dataset Construction:** In addition to the trace examples in our constructed VTS dataset, the trained models retain and generate similar multi-step reasoning traces during inference. These traces are observable in Figure 1 and in Appendix C.3, where models produce dynamic reasoning involving tool selection and verification-guided iteration. Each generation includes a full trajectory with tool actions and outputs.
>
> **3. Empirical Support via Multi-Step Trace Lengths and Tool Diversity:** Across multiple benchmarks (e.g., BLINK, V*Bench), our models often generate longer and more structured reasoning trajectories compared to vanilla baselines. For instance, average reasoning steps increase from ~1.5 (GPT-4o baseline) to ~3.2 (ours with VTS-V), and tool diversity is observed in the trace logs, including frequent use of grounding, depth, OCR and cropping actions.
>
> **4. Future Work on Manual Evaluations:** We agree that a manual or crowdworker-based evaluation of reasoning quality (e.g., alignment between trace and final answer) would further strengthen this claim. We plan to include such analysis in future work, but our current framework already outputs verifiable, human-readable sub-decisions, which is a step beyond traditional black-box VLM inference.
>
> We will revise our manuscript to clarify this distinction and to emphasize that the interpretability claim refers to the *explicit, grounded, and stepwise trace format* enabled by our model, rather than implying subjective human quality judgments alone.
>
> > **Response to W2: Relation to Existing Feedback-Based Reflective Mechanisms**
> >
>
> We thank the reviewer for pointing out the need to better clarify how our verifier differs from existing reflective or feedback-based mechanisms. We agree that our original related work discussion could have better distinguished the motivation and contributions of our verifier design.
>
> While prior works such as V-STaR and step-level math verifiers adopt reflective paradigms for single-domain problems (e.g., mathematical reasoning), our verifier is designed for multi-modal, tool-augmented, and multi-step visual reasoning. Most existing verifiers operate in textual domains, generate scalar scores or binary accept/reject signals, and are trained with task-specific preference data.
>
> Our method differs in the following key aspects:
>
> - We design a verifier that operates over multi-modal reasoning trajectories involving visual tools and structured actions, rather than purely textual rationales.
> - We train the verifier using multi-step preference comparisons derived from our large-scale and diverse VTS-DPO dataset, which is constructed from the LLaVA-OneVision corpus and covers over 80 real-world task types.
> - Our verifier is not just a reward model but actively controls reasoning flow during inference, enabling early stopping and action rejection.
>
> We will revise our related work section accordingly to better reflect these distinctions.
>
> > **Response to Q2: Clarifying the Contribution of the Framework vs. the Dataset**
> >
>
> Thanks for the thoughtful question. We agree that distinguishing between the impact of the framework (VTS-V) and the curated dataset (VTS-SFT) is important.
>
> Our experimental results (see table below, excerpted from Table 1) demonstrates that the framework alone leads to substantial improvements when applied to a capable model like GPT-4o *without any fine-tuning*:
>
> | **Model** | **Avg Accuracy (BLINK)** |
> | --- | --- |
> | GPT-4o | 62.25% |
> | GPT-4o + VTS-V | **69.15% (+6.9)** |
>
> This indicates that the VTS-V framework itself is effective, assuming the base model can understand multi-turn instructions and tool-augmented reasoning workflows.
>
> In Table 2, applying the same framework to open-source models without fine-tuning leads to performance degradation. This is not a limitation of the framework, but rather a reflection of the lack of tool-use priors and alignment tuning in these models. Specifically:
>
> - Open-source models have no pretraining exposure to structured reasoning formats like: Plan → Action → Tool → Observation → Continue/Stop.
> - They often misinterpret verifier signals or fail to produce valid tool-use outputs without explicit instruction.
>
> By contrast, GPT-4o benefits from strong instruction-following and alignment capabilities, allowing it to integrate seamlessly with VTS-V.
>
> After fine-tuning open-source models with VTS-SFT, they learn to properly use the framework and outperform their original baselines (e.g., Qwen2VL-7B improves from 48.01% to 52.19%).
>
> We will clarify this distinction in the revision.
>
> > **Response to Q3: Ablation Without the Reasoner**
> >
>
> Thanks for the suggestion. To directly evaluate the contribution of the **reasoner module**, we compare models with:
>
> - **w/o Reasoner**: The base model uses verifier guidance but no trained reasoner (tool actions are generated directly).
> - **w/ Reasoner**: Our full VTS-V framework, with both trained reasoner and verifier.
>
> We report the results below:
>
> | Model Variant | V*Bench | MMStar | MathVista |
> | --- | --- | --- | --- |
> | Qwen2.5VL-7B + Verifier (w/o R) | 67.54 | 55.93 | 7.80 |
> | Qwen2.5VL-7B + VTS-V (w/ R) | 75.13 | 57.93 | 23.52 |
> | Qwen2VL-7B + Verifier (w/o R) | 50.80 | 44.29 | 20.24 |
> | Qwen2VL-7B + VTS-V (w/ R) | 66.67 | 55.26 | 23.80 |
>
> The results demonstrate that incorporating the trained **reasoner** significantly improves performance across all benchmarks. Without the reasoner (Verifier-only), models struggle to perform reliable multi-step reasoning—especially on complex tasks like MathVista. Adding the reasoner consistently leads to large gains (e.g., +15.72 on MathVista for Qwen2.5VL-7B), highlighting its importance in executing valid tool-use actions and following verifier guidance effectively.
>
> We will highlight this finding more clearly in the revised version.
>
> > **Response to AQ1 & AQ2: Framework Visualization and Appendix Formatting**
> >
>
> Thanks for the feedback. We have added improved visualizations of the framework and corrected the formatting issues in the appendix. These updates will be reflected in the revised version of the paper.

---

> > ### Comment · Reviewer_XMPf · 2025-08-04
> >
> > Thanks for the authors' response and addition experiments to address my concerns. As they provide strong evidence supporting the effectiveness of the Reasoner and their proposed framework. I also appreciate the responses regarding w2, and I believe a more detailed comparison is necessary in revised version, even only appear in the Appendix. Although I maintain my concerns regarding the manual evaluation, which is not fully addressed, I will still increase my rating from 3 to 4 because authors provide persuasive technical clarification for most of my concerns.
> > Hope to see updated discussions in the final revision.

---

> > > ### Author Response · Authors · 2025-08-04
> > > **Thanks for your reply!**
> > >
> > > Thank you for your thoughtful feedback and for raising your rating. We sincerely appreciate your time and constructive suggestions, and we will fully address your remaining concerns in the final revision. Specifically, we will provide a more comprehensive comparison of our verifier versus prior reflective mechanisms (W2) and include additional discussion about manual evaluation results demonstrating our method's interpretability.

---

> ### Comment · Area_Chair_NFXJ · 2025-08-03
>
> Dear Reviewer,
>
> Could you please check if the authors’ rebuttal adequately addresses your concerns? If so, kindly acknowledge the rebuttal and provide any additional comments. If not, it would be greatly appreciated it if you could engage in a discussion with the authors. Your input at this stage is essential to the review process. Thank you very much for your time and effort!
>
> AC

---

> ### Comment · Area_Chair_NFXJ · 2025-08-06
>
> Dear Reviewer,
>
> Please remember to submitting the *Mandatory Acknowledgement*. Thank you!
>
> AC

---

### Official Review · Reviewer_weSb · 2025-07-03

**Clarity:** 3
**Significance:** 2
**Originality:** 2
**Rating:** 4
**Confidence:** 4

**Summary:**

This paper addresses the limitation of conventional multimodal large language models (MLLMs) with fixed and static visual tokens input, hindering iterative refinement of visual understanding. To bridge this gap, the authors propose a framework named Visual Token Scaling with Verification (VTS-V), which enables MLLMs to perform iterative, verifier-guided reasoning over visual content.

VTS-V formulates visual reasoning as a Markov Decision Process (MDP) consisting of two core components: a reasoner that generates visual reasoning steps (planning, action selection, and observation) and a verifier trained via multi-step Direct Preference Optimization (DPO) to evaluate action quality and determine termination. To support training, the authors introduce the VTS dataset, comprising VTS-SFT (315K supervised reasoning trajectories with tool use) and VTS-DPO (301K preference-labeled trajectory pairs).

Experimental results across diverse benchmarks (BLINK, V*Bench, MMStar, MathVista) demonstrate that VTS-V significantly outperforms baselines. For closed-source models like GPT-4o, it improves average accuracy on BLINK from 62.25% to 69.15%. For open-source models (e.g., Qwen2-VL-7B, LLaMA3.2-11B), fine-tuning on VTS-SFT followed by VTS-V yields gains of 4.18% to 7.58%. Ablation studies confirm the verifier’s critical role in filtering suboptimal paths.

**Questions:**

See the Weaknesses section for major concerns. Besides, I would like to ask the following questions:

1. Since the pipeline is modeled as Markov Decision Process, why not train the models use RL algorithms (e.g., GRPO)?
2. What if we train a single model to perform reasoning actions and verification both, instead of two separate models? Would it be more efficient?
3. Can we use the trained VTS-V pipeline to construct a new iteration of VTS dataset, and iteratively boosting the framework in a self-teaching manner?

**Ethical Concerns:**

["NO or VERY MINOR ethics concerns only"]

**Final Justification:**

Thank you to the Authors for your patient rebuttal; my concerns have now largely been addressed. Based on this work's contributions to dataset construction and the use of verifiers, I believe it deserves to be accepted.​

**Limitations:**

yes

**Quality:**

3

**Strengths And Weaknesses:**

**Strengths:**
- VTS-V effectively addresses the static inference limitation of MLLMs by introducing dynamic visual token scaling and verifier-guided iteration, aligning more closely with human-like dynamic perception.
- The VTS dataset (VTS-SFT and VTS-DPO) fills a critical gap in tool-augmented visual reasoning data, providing high-quality supervised and preference-labeled trajectories to train both the reasoner and verifier.
- Experiments span diverse benchmarks and model types (closed-source and open-source), showing consistent improvements in accuracy and interpretability.

**Weaknesses:**
- Although the work is solid, the method is more like a combination of previous works [1] (for iterative observation and reasoning), [2] (for reasoner and verifier separation) and [3] (for tool-use), thus its novelty is limited.
- The paper highlights "visual token scaling" as a key innovation, but it fails to clearly exhibit the scaling mechanism, e.g., a curve demonstrating the scaling law for visual tokens.
- The verifier’s performance depends heavily on the VTS-DPO dataset, and its ability to adapt to unseen tasks or domains is not thoroughly explored. This raises questions about its robustness in real-world, diverse environments.

[1] Penghao Wu and Saining Xie. $V^*$: Guided visual search as a core mechanism in multimodal llms. In Proceedings of the IEEE/CVF Conference on Computer Vision and Pattern Recognition, pp. 13084–13094, 2024.

[2] ArianHosseini,XingdiYuan,NikolayMalkin,AaronCourville,AlessandroSordoni,and RishabhAgarwal. V-star: Training verifiers for self-taught reasoners. arXiv preprint arXiv:2402.06457, 2024.

[3 ]Yushi Hu, Otilia Stretcu, Chun-Ta Lu, Krishnamurthy Viswanathan, Kenji Hata, Enming Luo, Ranjay Krishna, and Ariel Fuxman. Visual program distillation: Distilling tools and programmatic reasoning into vision-language models. In Proceedings of the IEEE/CVF Conference on Computer Vision and Pattern Recognition, pp. 9590–9601, 2024.

---

> ### Author Rebuttal · Authors · 2025-07-31
>
> **Overall Response**
>
> We appreciate the thoughtful and constructive feedback! In response, we clarified the novelty of our verifier-guided MDP framework beyond prior work, provided visual token scaling analyses, and demonstrated strong generalization of the verifier across unseen domains. We also addressed design choices such as reasoner-verifier separation, the use of SFT over RL, and potential future directions like self-bootstrapping. These clarifications and results will be incorporated into the final version if the paper is accepted.
>
> > **Response to W1: Novelty Beyond Combining Prior Work**
> >
>
> We appreciate this thoughtful feedback. While our work draws inspiration from prior efforts in iterative reasoning [1], reasoner-verifier separation [2], and tool use [3], we believe our contributions go beyond a simple combination. Specifically, we introduce a principled, theoretically-grounded framework tailored for multi-step visual reasoning with tool use, supported by both new methodology and new data.
>
> Key distinctions:
>
> - **MDP-based Reasoning Framework:**  Unlike V*[1], which focuses on task-specific cropping, we formulate reasoning as a Markov Decision Process over a diverse toolset (OCR, grounding, depth, etc.). This allows for flexible, dynamic visual token scaling across tasks and models.
>
> - **Verifier for Guided Inference with Theoretical Guarantees:**  Compared to V-STaR [2], which ranks final answers, our verifier operates at each step, providing real-time guidance and bounded termination (Theorem 3.2), enabling interpretable and efficient inference.
>
> - **New Dataset for Multi-Step Visual Tool Use:**  Prior tool-use works [3] do not support multi-turn visual reasoning. We contribute the VTS-SFT and VTS-DPO datasets, the first to support fine-grained tool selection and verifier training over >80 diverse tasks.
>
>
> Together, our framework enables strong generalization (e.g., +6.9% on GPT-4o without fine-tuning) and interpretability, establishing a solid foundation for future multi-modal reasoning systems. We will revise the manuscript to more clearly highlight these distinctions.
>
> > **Response to W2:  Visual Token Scaling Mechanism with Optimal-Stopping Verifier**
> >
>
> We sincerely appreciate the reviewer’s insightful feedback regarding the need to clarify the visual token scaling mechanism. Below, we present selected experimental results demonstrating the scaling relationship between multi-step reasoning performance and output visual tokens, which reflects our framework’s dynamic behavior.
>
> - Results based on closed-source model APIs:
>
> | **Visual & Text Tokens Per Sample** | **GPT-4o BLINK Counting Accuracy** |
> | --- | --- |
> | 550 | 49.17% |
> | 1100 | 58.33% |
> | 1650 | **67.50%*** |
> | 2200 | 64.16% |
> | 2750 | 65.83% |
>
> - Results for the trained open-source models:
>
> | **Visual & Text Tokens Per Sample** | **Qwen2.5-VL BLINK Fun.Corr.** | **Qwen2-VL BLINK Depth** | **LLaMA-3.2-Vision BLINK Spatial** |
> | --- | --- | --- | --- |
> | 600 | 19.23% | 56.45% | 62.94% |
> | 1200 | 23.85% | **60.48%*** | 65.03% |
> | 1800 | 25.38% | 58.87% | **69.23%*** |
> | 2400 | **30.00%*** | 57.26% | 67.83% |
> | 3000 | 26.15% | 58.06% | 67.13% |
>
> The tables above illustrate the relationship between reasoning accuracy and visual token usage for both GPT-4o and our trained models. On average, GPT-4o and our trained models generate approximately 50 and 100 text tokens per reasoning step, respectively, while processing ~100 visual tokens through dynamic image operations. We control the number of visual tokens by limiting the reasoner’s maximum steps, while our verifier selects the optimal stopping point (marked by *).
>
> Key findings:
> - Non-monotonic scaling: Accuracy improves with more visual tokens initially, but may decline if reasoning becomes unnecessarily long.
> - Verifier-guided optimality: Our VTS-V verifier consistently identifies the best stopping point, balancing depth and efficiency.
> - Robust reasoning: Even beyond the optimal step, accuracy remains higher than with under-reasoning, highlighting the benefits of sufficient visual grounding.
>
> We will incorporate these scaling trends and supporting visualizations into the revised manuscript.
>
> > **Response to W3: Robustness and Generalization of the Verifier Trained on VTS-DPO**
> >
>
> We thank the reviewer for raising concerns regarding the verifier’s dependence on the VTS-DPO dataset and its robustness to unseen domains. We provide further clarification below.
>
> The VTS-DPO dataset is constructed based on the LLaVA-OneVision (LLaVA-OV) dataset, which is currently among the most comprehensive multi-modal corpora, covering over 80 distinct vision-language task types. Our DPO dataset comprises 301,028 preference pairs, carefully curated to span a wide range of visual reasoning domains. These include document understanding, scientific diagrams, mathematical problem solving, chart QA, OCR, and general commonsense VQA.
>
> The dataset is divided into the following major categories:
>
> | **Domain Category** | **# Tasks** | **# Examples** |
> | --- | --- | --- |
> | General | 25 | 135,804 |
> | Document/Chart/Screen | 18 | 100,681 |
> | Math/Logical Reasoning | 15 | 52,049 |
> | OCR/Text-grounded Tasks | 6 | 12,494 |
>
> Each category includes examples from highly varied sources.
>
> Notably, all benchmarks used for evaluation in our experiments (e.g., BLINK, V*Bench, MMStar, MathVista) are held-out from the training data. The verifier is therefore evaluated on unseen reasoning tasks and domains. The strong performance observed across these evaluations supports the verifier’s ability to generalize beyond the training distribution.
>
> We will expand on the construction and coverage of VTS-DPO in the revised Appendix, including a full breakdown of dataset sources and domains.
>
> > **Response to Q1: Why not use Reinforcement Learning (e.g., GRPO) to Train the Reasoner?**
> >
>
> We appreciate the reviewer’s suggestion to explore GRPO for training the reasoner. We have conducted preliminary experiments and found that GRPO performs poorly in our multi-tool setting, where the reasoning space is large and tools are diverse. While prior works using GRPO (e.g., DeepEyes [1]) focus on limited-tool environments.
>
> To address this, we adopt supervised fine-tuning (SFT) as a more stable and effective initialization strategy. Our VTS-SFT dataset serves as a strong cold-start foundation for future RL-based methods. While GRPO is not the focus of this paper, we are actively exploring its potential in future work with improved reward design and tool selection policies.
>
> [1] Zheng Z, Yang M, Hong J, et al. *DeepEyes: Incentivizing “Thinking with Images” via Reinforcement Learning*. arXiv preprint arXiv:2505.14362, 2025.
>
> > **Response to Q2: Why Separate the Reasoner and Verifier Instead of Using a Single Unified Model?**
> >
>
> We thank the reviewer for this thoughtful question. In principle, combining reasoning and verification into a single model is an appealing idea for potential efficiency. However, based on both theoretical motivations and empirical observations, we find that the decoupled design is more effective in practice, especially in the context of multi-modal, tool-augmented reasoning.
>
> **1. Current Models Struggle to Jointly Handle Both Tasks Well:** Reasoning and verification represent distinct capabilities: the reasoner must generate tool-using action plans conditioned on complex context, while the verifier needs to evaluate the overall quality or utility of each reasoning step. In practice, we observe that when both responsibilities are handled by a single model, the performance on one or both tasks deteriorates.
>
> **2. GPT-4o Results Support the Effectiveness of Separation:** Our experimental results (see table below, excerpted from Table 1) show that adding a standalone verifier to GPT-4o leads to significant performance improvements:
>
> | **Model** | **Avg Accuracy (BLINK)** |
> | --- | --- |
> | GPT-4o | 62.25% |
> | GPT-4o + Verifier | **69.15%** (+6.9) |
>
> This demonstrates that adding a verifier as a separate module to an already-strong base model (GPT-4o) yields substantial gains, especially on tasks that require structured multi-step reasoning (e.g., +18.33 on Counting, +12.91 on Functionally-Correlated tasks).
>
> **3. Modularity and Reusability of Verifier:** Our verifier can be viewed as a plug-in component: it is model-agnostic and can be reused across architectures (e.g., GPT-4o, Qwen, LLaMA) once trained. This improves flexibility, reduces fine-tuning costs, and makes the framework extensible to new tasks or domains.
>
> We will clarify this design decision in the final version of the paper.
>
> > **Response to Q3: Iterative Bootstrapping of VTS via Self-Teaching**
> >
>
> We appreciate the reviewer’s insightful suggestion.
>
> Using the trained VTS-V pipeline to construct additional VTS-style data in a self-bootstrapping fashion is indeed an interesting and promising direction. It aligns well with recent trends in self-improving agents and iterative data refinement.
>
> One potential challenge, however, lies in maintaining supervision quality during self-generated data collection. Our VTS-SFT and VTS-DPO datasets rely on high-quality reasoning trajectories and carefully curated preference signals. If the model-generated traces contain subtle reasoning errors or invalid tool calls, these could accumulate and introduce bias or instability during future training rounds.
>
> That said, combining model-generated traces with selective human or verifier-based filtering may offer a scalable way forward. We see this as an exciting avenue for future work and will consider incorporating this direction into our ongoing research agenda.

---

> > ### Comment · Reviewer_weSb · 2025-08-03
> >
> > Thank you to the Authors for your patient rebuttal; my concerns have now largely been addressed. Based on this work's contributions to dataset construction and the use of verifiers, I believe it deserves to be accepted.​

---

> > > ### Author Response · Authors · 2025-08-03
> > > **Thanks for your reply!**
> > >
> > > Thank you for your thoughtful review and constructive feedback on our paper. We sincerely appreciate your time and effort in evaluating our work, and we are delighted that our rebuttal addressed your concerns. We will carefully incorporate your valuable suggestions to further improve our manuscript in the final version.

---

> ### Comment · Area_Chair_NFXJ · 2025-08-03
>
> Dear Reviewer,
>
> Could you please check if the authors’ rebuttal adequately addresses your concerns? If so, kindly acknowledge the rebuttal and provide any additional comments. If not, it would be greatly appreciated it if you could engage in a discussion with the authors. Your input at this stage is essential to the review process. Thank you very much for your time and effort!
>
> AC

---

### Official Review · Reviewer_MBvX · 2025-07-03

**Clarity:** 3
**Significance:** 3
**Originality:** 3
**Rating:** 5
**Confidence:** 4

**Summary:**

This paper presents VTS-V, a novel framework for inference-time visual token scaling in multi-modal large language models (MLLMs). Inspired by the dynamic, feedback-driven nature of human visual perception, the authors model visual reasoning as a Markov Decision Process. The framework consists of a reasoner that plans visual actions and a verifier trained via multi-step Direct Preference Optimization (DPO) to guide the reasoning steps and determine termination. To support this framework, the authors introduce VTS, a new dataset including supervised reasoning trajectories (VTS-SFT) and preference-labeled trajectory pairs (VTS-DPO). They demonstrate consistent improvements over several benchmarks, showing enhanced performance and interpretability.

**Questions:**

Please address the weaknesses above.

**Ethical Concerns:**

["NO or VERY MINOR ethics concerns only"]

**Final Justification:**

The well-designed iterative pipeline seems to be practical for visual token scaling, and the proposed dataset is useful for fine-tuning open-sourced models. I believe this work is deserved to be accepted.

**Limitations:**

yes

**Quality:**

3

**Strengths And Weaknesses:**

Strengths:
- The iterative inference-time scaling of visual tokens using verifier-guided reasoning presents a new perspective in the field.
- The iterative nature of VTS-V provides clear, interpretable reasoning paths, enhancing the transparency of model decisions.
- The new dataset (VTS) provides valuable resources for further research.

Weaknesses:
- The complexity of implementation and computation inherent in training and inference procedures may limit widespread adoption. I encourage the authors to include a more detailed analysis of inference-time overhead, such as latency, and memory usage across models and datasets.
- While the authors present comprehensive results for GPT-4o variants on the BLINK benchmark, the evaluation on other datasets such as V*Bench, MMStar, and MathVista is only conducted for open-source models. As GPT-4o represents a strong closed-source baseline, its exclusion from these additional benchmarks limits the understanding of how the proposed VTS-V framework generalizes across diverse reasoning tasks for top-tier models. Including these comparisons would strengthen the empirical claims.
- The paper observes that integrating VTS-V with GPT-4o yields consistent improvements, even without additional fine-tuning, whereas applying the same method to open-source models (e.g., Qwen2-VL, LLaMA3.2) without fine-tuning leads to significant performance drops. However, this discrepancy is not thoroughly analyzed. The authors are encouraged to investigate and discuss potential reasons behind this behavior—for example, differences in instruction-following ability, pretraining data coverage, or inherent robustness to tool-augmented workflows.

---

> ### Author Rebuttal · Authors · 2025-07-31
>
> **Overall Response**
>
> Thanks for the detailed and thoughtful feedback. In response, we provided a quantitative analysis of inference-time latency and memory overhead, demonstrating the efficiency–accuracy tradeoffs of our verifier-guided framework. We also expanded GPT-4o evaluations across V*Bench, MMStar, and MathVista, showing strong gains over both baselines and alternative prompting methods. Finally, we clarified why our framework improves GPT-4o without fine-tuning but requires VTS-SFT to enable open-source models to follow structured, tool-based reasoning. All clarifications and new results will be incorporated into the revised version if the paper is accepted.
>
> > **Response to Weakness 1: Inference-time Overhead (Latency & Memory Usage)**
> >
>
> We appreciate the reviewer’s thoughtful concern regarding inference complexity. To address this, we now provide a detailed quantitative analysis of inference-time overhead, including latency, memory usage, and the number of intermediate tokens generated during dynamic reasoning.
>
> Our method introduces iterative steps composed of:
>
> - Verifier calls, which require only prefill computation (i.e., forward pass over the input context to generate a stop/continue decision),
> - Reasoner calls, which involve autoregressive decoding to produce tool instructions.
>
> Each interaction is structured as:
>
> ```jsx
> context = input_prompt: Q (image, text_question)
> while verifier(context) == 'continue':
> 	current_output = reasoner(context)
> 	physical_feedback = tool_using_API(current_output)
> ```
>
> We define the total inference latency as:
>
> $$\text{Latency} = \sum\_{i=1}^{H} \left( \frac{C\_i}{P\_t} + \frac{O\_i}{D\_t} \right)$$
>
> Where:
>
> - $H$: total number of reasoning steps;
> - $C\_i$: number of input tokens at step i (verifier prefill);
> - $O\_i$: number of generated tokens at step i (reasoner decode);
> - $P\_t$: model’s prefill throughput (tokens/sec);
> - $D\_t$: model’s decode throughput (tokens/sec).
>
> **Example Overhead (Qwen2.5-VL-7B-Instruct with VTS-V):**
>
> - Average steps (H): 3.1
> - Average verifier prefill tokens: ~10058.0 per step
> - Average reasoner decode tokens: ~95.3 per step
> - Throughput on 80GB A800 GPUs (float16):
>     - $P\_t \approx$ 18493.6 tokens/s
>     - $D\_t \approx$ 231.8 tokens/s
>
> **Total latency ≈ 3.1 × (10058.0 / 18493.6 + 95.3 / 231.8) ≈ 2.96 seconds**
>
> **Memory Usage:**
>
> - With max_model_len=65536 and dtype=float16, our models can be deployed within 32GB through vLLM or SGLang on 80GB A800 GPUs.
> - All tool models (including GroundingDINO, Depth-Anything, etc.) collectively require approximately 23 GB of additional memory during inference.
>
> **Tradeoffs and Mitigations:**
>
> - We find a clear tradeoff between reasoning depth and computational cost, which is justified by significant accuracy gains (e.g., +7.58% avg. on BLINK for Qwen2.5-VL-7B-Instruct).
> - To mitigate latency for deployment, our framework allows *parallel verifier batching*, *early stopping*, and *adaptive step truncation* (via ε-threshold).
>
> We will include this analysis, along with latency and memory tables across models, in the revised Appendix B. Thank you again for the helpful suggestion.
>
> > **Response to W2: GPT-4o Evaluation on Broader Benchmarks**
> >
>
> We sincerely appreciate the reviewer’s insightful feedback regarding the evaluation of our VTS-V framework on additional benchmarks for GPT-4o. We agree that including GPT-4o results on V*Bench, MMStar, and MathVista would better demonstrate the framework’s generalization across diverse reasoning tasks.
>
> To address this, we conducted additional evaluation experiments, including GPT-4o, GPT-4o + CoT[1] and GPT-4o + Visual Sketchpad [2] as baselines. Our results are summarized below:
>
> |  |  V*Bench | MMStar | MathVista |
> | --- | --- | --- | --- |
> | GPT-4o | 60.73 |  58.40 | 63.80 |
> | GPT-4o + CoT | 60.73 | 56.47 | **66.50** |
> | GPT-4o + Sketchpad | 71.20 | 60.33 | 59.40 |
> | GPT-4o + VTS-V | **73.82** | **65.80** | 65.90 |
>
> Key observations:
>
> - Our VTS-V achieved state-of-the-art performance on both V*Bench and MMStar, with absolute improvements of **13.09%** and **7.4%** over the baseline GPT-4o, respectively,  demonstrating the efficiency of our approach.
> - Our method outperformed GPT-4o by 2.1% on MathVista. And it achieved nearly equivalent performance to the top-performing GPT-4o+CoT (a mere 0.6% difference), further demonstrating the competitiveness of our approach.
>
> [1] Wei J, Wang X, Schuurmans D, et al. Chain-of-thought prompting elicits reasoning in large language models[J]. Advances in neural information processing systems, 2022, 35: 24824-24837.
>
> [2] Hu Y, Shi W, Fu X, et al. Visual sketchpad: Sketching as a visual chain of thought for multimodal language models[J]. Advances in Neural Information Processing Systems, 2024, 37: 139348-139379.
>
> > **Response to W3: Performance Gap Between GPT-4o and Open-Source Models Without Fine-Tuning when integrating VTS-V**
> >
>
> We thank the reviewer for raising this important point. While Section 5.2 (lines 306–317) of the main paper briefly mentions this observation, we now provide a more thorough explanation of why VTS-V improves performance in GPT-4o even without fine-tuning, but degrades performance in open-source models like Qwen2-VL and LLaMA-3.2-Vision.
>
> Applying VTS-V directly to GPT-4o yields consistent improvements across benchmarks without any additional training. In contrast, applying the same framework to open-source models *without fine-tuning* leads to degraded performance.
>
> We hypothesize that this performance discrepancy stems from the following factors:
>
> 1. **Lack of Tool-Using Priors in Open-Source Models.**
>     Unlike GPT-4o, open-source models are not pretrained or aligned with tool-augmented workflows. They have no exposure to structured reasoning formats like: Plan → Action → Tool call → Observation → Continue/Stop. As a result, when presented with verifier signals and expected to issue tool-use commands, these models either ignore the prompt or generate malformed instructions.
>
> 2. **Robust Instruction Following in GPT-4o.**
>     GPT-4o has undergone extensive alignment tuning (e.g., RLHF, tool-use demonstrations) which enables it to robustly interpret multi-turn instructions and unfamiliar prompting formats. This generalization allows it to cooperate with our verifier module out of the box, without dedicated SFT.
>
> 3. **Verifier Signal Interpretation.**
>     Verifier guidance in our framework (e.g., continue vs. stop) depends on implicit task understanding and adaptive response. GPT-4o can interpret these signals semantically, while open-source models require fine-tuning to learn how to react to verifier outputs correctly.
>
> 4. **Empirical Confirmation via SFT.**
>     After supervised fine-tuning on our VTS-SFT dataset, open-source models *learn valid tool-use behaviors* and align well with verifier guidance. Their performance not only recovers but exceeds the original baseline (e.g., Qwen2VL-7B-Instruct improves from 48.01% to 52.19%).
>
>
> We will clarify this distinction and analysis in the revision. This behavior gap also underscores the importance of our VTS-SFT dataset in enabling open-source models to benefit from verifier-guided multi-step reasoning workflows.

---

> > ### Comment · Reviewer_MBvX · 2025-08-04
> >
> > Thank you to the Authors for your rebuttal; my concerns have now largely been addressed. I'll keep my score.

---

> > > ### Author Response · Authors · 2025-08-04
> > > **Thanks for your reply!**
> > >
> > > Thank you for your time and valuable feedback. We sincerely appreciate your effort in reviewing our work and are grateful for your positive assessment. Your insights have helped strengthen the paper, and we will incorporate them to further refine the final manuscript.

---

> ### Comment · Area_Chair_NFXJ · 2025-08-03
>
> Dear Reviewer,
>
> Could you please check if the authors’ rebuttal adequately addresses your concerns? If so, kindly acknowledge the rebuttal and provide any additional comments. If not, it would be greatly appreciated it if you could engage in a discussion with the authors. Your input at this stage is essential to the review process. Thank you very much for your time and effort!
>
> AC

---

> ### Comment · Area_Chair_NFXJ · 2025-08-06
>
> Dear Reviewer,
>
> Please remember to submitting the *Mandatory Acknowledgement*. Thank you!
>
> AC

---

### Note · Authors · 2025-08-14

We sincerely appreciate the reviewers’ thoughtful feedback and constructive critiques.

### **Key strengths highlighted by reviewers**

- **Principled and novel framework:** [*Reviewer MBvX*, *Reviewer weSb*] praised our verifier-guided, MDP-based design with clear theoretical grounding and its novelty for dynamic multi-step visual reasoning.
- **Valuable dataset:** [*Reviewer MBvX*, *Reviewer weSb*, *Reviewer XMPf*, *Reviewer CE1C*] recognized the VTS dataset (VTS-SFT & VTS-DPO) as a significant resource for reasoner–verifier training across diverse tasks.
- **Strong empirical performance:** [*Reviewer weSb*] noted our consistent accuracy gains across multiple benchmarks for both closed- and open-source models.
- **Interpretability and applicability:** [*Reviewer XMPf*, *Reviewer CE1C*] valued our explicit, grounded reasoning traces, the effectiveness of the reasoner–verifier design, and the relevance of our method to complex visual reasoning.

### **Addressing reviewer concerns**

We have carefully revised the manuscript to resolve the following concerns:

- **Inference-time overhead:** [*Reviewer MBvX*] We added quantitative latency/memory analysis and proposed optimizations such as parallel verifier batching and adaptive step truncation.
- **Broader GPT-4o evaluation:** [*Reviewer MBvX*] We added results on V*Bench, MMStar, and MathVista, showing substantial gains over baselines.
- **Performance gap explanation:** [*Reviewer MBvX*, *Reviewer XMPf*] We analyzed differences in tool-use priors and instruction-following ability; showed VTS-SFT fine-tuning enables open-source models to fully benefit.
- **Verifier robustness:** [*Reviewer weSb*] We detailed VTS-DPO dataset diversity and demonstrated generalization to unseen benchmarks.
- **Interpretability evidence:** [*Reviewer XMPf*] We refined our definition of interpretability and added ablations showing the necessity of the reasoner.
- **Novelty, scaling, and distinctions:** [*Reviewer weSb*, *Reviewer CE1C*] We clarified differences from prior work, presented token–accuracy scaling trends, demonstrated verifier-guided optimal stopping, and distinguished our method from multi-round RL and dynamic perception approaches.

These revisions were well-received, with multiple reviewers confirming their concerns were addressed and some increasing their scores. We believe these revisions comprehensively address the reviewers’ concerns while strengthening the paper’s contributions and accessibility.

---

### Decision · Program_Chairs · 2025-09-17

**Decision:**

Accept (poster)

**Comment:**

This paper introduces an inference-time visual token scaling approach by modeling visual reasoning as a Markov Decision Process with two main components: a reasoner and a verifier. The authors also present a dataset with supervised reasoning trajectories incorporating tool use for training purposes. Experimental results demonstrate consistent improvements across several benchmarks.

The reviewers agreed that the verifier-guided reasoning is a unique contribution. They also noted that the proposed dataset has strong potential to inspire future work in the field. While reviewers initially raised concerns about novelty, training details, and the need for additional analysis, these were well addressed during the rebuttal phase and confirmed by the reviewers.

The AC agrees with the reviewers’ recommendation of acceptance.